# Exposing the Achilles' Heel:
# Evaluating LLMs Ability to Handle Mistakes in Mathematical Reasoning

## Abstract

Large Language Models (LLMs) have significantly impacted the field of Math Word Problems (MWPs), transforming how these problems are approached and solved, particularly in educational contexts. However, existing evaluations often focus on final accuracy, neglecting the critical aspect of reasoning capabilities. This work addresses that gap by evaluating LLMs' abilities to detect and correct reasoning mistakes. We present a novel dataset, MWP-MISTAKE, containing MWPs with both correct and incorrect reasoning steps generated through rule-based methods and smaller language models. Our comprehensive benchmarking of state-of-the-art models such as GPT-4o and GPT4 uncovers important insights into their strengths and limitations. While GPT-4o excels in mistake detection and rectification, gaps remain, particularly in handling complex datasets and novel problems. Additionally, we identify concerns with data contamination and memorization, which affect LLM reliability in real-world applications. While OpenAI's O1 model demonstrates 90% accuracy in reasoning and final answers on complex tasks, it remains weak in mistake detection. Our findings highlight the need for improved reasoning evaluations and suggest ways to enhance LLM generalization and robustness in math problem-solving.

## 1 Introduction

Large Language Models (LLMs) have transformed artificial intelligence applications across diverse domains, including healthcare, agriculture, and education (OpenAI, b;a). Their remarkable capabilities in natural language understanding, question answering, and mathematical problem-have shown potential to revolutionize various human endeavors (Liu et al., 2024b). Recent advancements have fueled extensive research into applying LLMs to interpret and solve a wide array of mathematical tasks, from basic arithmetic to complex algebraic equations and calculus problems (Hendrycks et al., 2021; Zhang et al., 2024a). Math Word Problems (MWPs) involve interpreting narrative scenarios to extract mathematical concepts and apply reasoning for solutions (Srivatsa & Kochmar, 2024). Studies (Xu et al., 2024; He-Yueya et al., 2023; Deb et al., 2023) show LLMs can convert text into mathematical expressions and generate accurate results, but a critical element *mathematical reasoning* is often underexplored.

Despite achieving remarkable accuracy rates exceeding 90% on datasets like GSM-8K (Grade School Math dataset with linguistically diverse word problems) (Cobbe et al., 2021a), foundational LLMs such as Claude-3-Opus (noa, a), Gemini Ultra (Team et al., 2024), and OpenAI GPT-4 (OpenAI et al., 2024) reveal a significant gap in our understanding of their capabilities in mathematical reasoning (Deb et al., 2023). Current research predominantly focuses on evaluating the final accuracy of MWPs (Luo et al., 2023; Yu et al., 2024), neglecting the intricate reasoning processes necessary to derive solutions. We argue that the reasoning steps play a pivotal role, and it is imperative to assess them to comprehensively analyze the foundational capabilities of these models. This necessity is further underscored by the increasing utilization of LLMs in domains such as education (Gan et al., 2023), where they serve as personalized tutors for students, aiding in teaching concepts and solving mathematical problems. Simply deriving the final answer is insufficient; the ability to guide students through correct steps, identify errors in their reasoning, and provide corrective guidance is paramount for such applications.

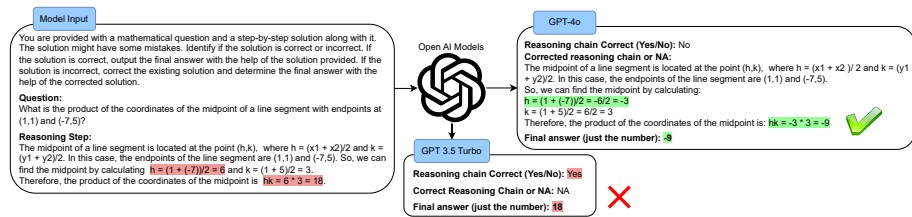

Figure 1: Model is prompted with a question along with incorrect reasoning steps to detect any mistake and correct the reasoning step to get to the correct final answer. GPT-4o generates the correct output, while GPT-3.5Turbo fails to identify any mistake in the reasoning step. (Task - T1)

This paper aims to bridge this gap by providing a comprehensive benchmark and evaluation of LLMs' performance on math word problems, including their capabilities in mistake detection and correction within the reasoning steps (Figure 1). Analyzing LLMs' ability to detect and rectify errors along the reasoning steps yields valuable insights into their overall problem-solving capabilities. Our objectives are threefold: firstly, to comprehensively evaluate LLMs' capabilities in mathematical reasoning, with a particular emphasis on mistake detection and correction; secondly, to identify the specific strengths and weaknesses of these models in handling various types of mathematical challenges; and thirdly, to propose potential directions for enhancing LLM capabilities in this domain.

To achieve this comprehensive evaluation, we have developed our own mistake dataset, designed to include errors in the reasoning steps. This dataset allows the assessment of models' proficiency not only in providing correct solutions but also in detecting and correcting mistakes within the reasoning steps. We evaluate 12 different foundational models including large, small and fine-tuned on math, language models on our curated dataset `MWP-MISTAKE`. We are releasing this dataset for further evaluation and benchmarking[1].

Our analysis reveals several key insights into the performance of LLMs on MWPs. Firstly, detecting mistakes, even trivial ones remains a significant challenge for these models. Secondly, LLMs often derive correct answers despite this difficulty in mistake detection. This can be attributed to data memorization and potential contamination in training datasets, where models may have encountered similar/same problems before. However, the ability to recover from or correct errors in the reasoning process is generally poor across most models. Our contributions to this paper are as follows:

1. We collect and release to the research community `MWP-MISTAKE`, a dataset containing MWPs with both correct and incorrect reasoning obtained from state-of-the-art MWP datasets such as SVAMP (Patel et al., 2021), GSM-8K (Cobbe et al., 2021b), MATH (Hendrycks et al., 2021), MATHBENCH (Liu et al., 2024a), and JEEBENCH (Arora et al., 2023). Incorrect reasoning is derived through meticulously crafted rules to alter the reasoning steps and using smaller models, leveraging their inherent limitations in solving MWPs.

2. We provide benchmark results for our dataset to evaluate the reasoning capabilities of LLMs such as GPT-4o (OpenAI, a), GPT-4 (OpenAI et al., 2024), GPT-3.5Turbo (noa, b), Claude (noa, a), as well as smaller language models like Llama (Touvron et al., 2023), Phi (Abdin et al., 2024a), and Mixtral (Jiang et al., 2024) and also models fine-tuned on Math datasets. Our analysis demonstrates that all SOTA LLMs struggle with mistake detection and correction.

3. Through meticulous evaluation and comparison of different LLMs, we offer a detailed analysis of their strengths and weaknesses in handling mathematical reasoning tasks. We also provide early preliminary evaluations with OpenAI o1 models, which still does not excel in mistake detection.

## 2   MWP-MISTAKE DATASET

Most Math Word Problem (MWP) datasets provide math problems with final answers, occasionally including correct reasoning steps. To evaluate LLMs' ability to detect and correct errors, we created the `MWP-MISTAKE` dataset using five sources: SVAMP (Patel et al., 2021), GSM-8K (Cobbe et al., 2021b), MATH (Hendrycks et al., 2021), MATHBENCH (Liu et al., 2024a), and JEEBENCH (Arora et al., 2023), with MATHBENCH and JEEBENCH being more recent. These five datasets form the basis of the MWP-MISTAKE dataset, covering a wide range of complexities from middle, high school to college levels. While GSM-8K and MATH offer ground truth corect reasoning steps, the others do not. For those, we used GPT-4 to generate chain-of-thought reasoning steps, which were

---

[1]Anonymous repository for source code and dataset: `https://anonymous.4open.science/r/Exposing-the-Achille-Heel-1D11/`

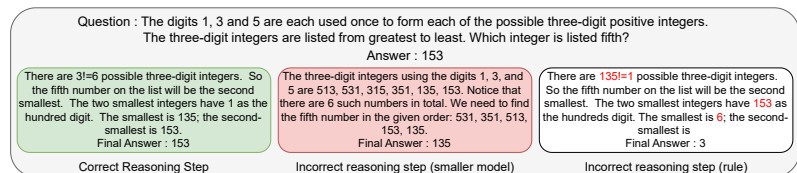

Figure 2: Examples of MWPs with correct reasoning, rule-based incorrect and smaller model based incorrect reasoning from MATH.

then extensively manually verified for correctness. The final dataset includes MWP questions, correct reasoning steps, and final answers from all five sources (see Appendix A for additional details). To create incorrect reasoning steps, we propose two approaches: (i) meticulously crafted rules, and (ii) using smaller models as bad reasoners, which we describe next.

## 2.1 METICULOUSLY CRAFTED RULES TO PROGRAMMATICALLY INJECT ERRORS

These rules are motivated and derived from common mistakes observed in educational settings, ensuring the errors introduced are realistic and representative of actual student errors.

1. **Shuffle numerical values:** Numerical values are shuffled among themselves to verify if models can correctly understand the question and select appropriate numerical values from the question.
2. **Replace numerical values:** Numerical values are replaced with random numbers ranging from 0 to 100. It identifies if the model can correctly pick the numerical values present in the question.
3. **Shuffle operations:** We randomly swap operators with other operators to test the model's ability to perform numerical operations.
4. **Insert random reasoning steps:** A random reasoning step is added at a random position to test the model's ability to identify incorrect reasoning.
5. **Shuffle reasoning steps:** The reasoning steps are shuffled to introduce ambiguity in the thought process. This tests whether the model can identify changes in reasoning order.
6. **Delete reasoning steps:** One reasoning step is deleted in solutions that have two or more steps. This helps to identify if the model can spot omissions in the reasoning process.

These rules mimic real-world student behavior by reflecting tendencies to get the order of steps wrong, skip steps, misinterpret numerical values, use incorrect numbers, apply the wrong mathematical operations, and add irrelevant steps in problem-solving. While rules #5 and #6 do not introduce explicit errors in reasoning, they are considered mistakes in our dataset to prompt the model to identify scenarios lacking clarity. Table 1 shows the number of questions selected from each of the five datasets to which these six rules are applied to curate incorrect reasoning. Thus, for every question selected, we created seven variations of reasoning steps (one correct + six incorrect).

## 2.2 SMALLER MODELS AS BAD REASONERS

Recently, SLMs are gaining popularity with increased performance, however, they still lack several capabilities, including advanced mathematical reasoning, resulting in poorer performance on MWPs. To curate incorrect reasoning steps, we use SLMs to generate Chain-of-Thought (COT) reasoning and final

Table 1: `MWP-MISTAKE` Dataset details with the total number of questions and reasoning steps.

| Dataset | Default reasoning | | Smaller model reasoning | | | |
|---|---|---|---|---|---|---|
| | # Questions with correct reason (GT) | # Questions with incorrect reason (Rules) | # Questions with incorrect reasoning | | | Total |
| | | | Llama-2-7b-chat | Mixtral-8x7B | Phi-3-mini | |
| SVAMP | 154 | 924 | 60 | 20 | 20 | 1178 |
| GSM-8K | 93 | 558 | 100 | 100 | 100 | 951 |
| MATH | 150 | 900 | 150 | 150 | 150 | 1500 |
| MATHBENCH | 100 | 600 | 100 | 100 | 100 | 1000 |
| JEEBENCH | 38 | 228 | 12 | 19 | 35 | 332 |

answers for all dataset questions. Questions with incorrect final answers, identified by comparing them to the ground truth, are retained, and their reasoning steps are classified as incorrect. We then perform an extensive human validation of the answer and reasoning steps to make sure their is a mistake (as there could few instances where the answer can be incorrect, but reasoning steps could be correct). We employ state-of-the-art SLMs, such as Llama-2-7b-chat, Phi-3-mini, and Mixtral-8x7B, to generate COT reasoning steps and Appendix C provides examples of such incorrect reasoning steps with final wrong answer. Table 1 provides statistics for each model across datasets. The entire dataset, including reasoning steps, was exhaustively manually verified to eliminate errors.

Our dataset includes questions with original correct reasoning, rule-based incorrect reasoning, and SLM-generated incorrect reasoning. For evaluation, we split the data into two parts: (1) **Default**, with correct reasoning and rule-based incorrect steps, and (2) **SLM reason**, featuring SLM-generated

incorrect reasoning. Table 1 provides the complete details of the curated `MWP-MISTAKE` dataset with the above two splits.

## 3 EXPERIMENTAL SETUP

**Task Details.** Our aim is to assess the performance of LLMs on MWPs, focusing on their ability to detect and correct mistakes within the reasoning steps. We have two task variants to accomplish this:

1. **Task-1 (T1):** Given a question and its reasoning steps, the model must identify correctness, rectify mistakes if needed and compute the final answer.(Figure 1).
2. **Task-2 (T2):** The model only needs to identify whether the reasoning steps provided are correct or incorrect and provide the final answer. No correction of reasoning steps is explicitly required.

In essence, T1 evaluates the model's ability to detect mistakes, rectify them, and derive the correct answer, while T2 focuses solely on detecting mistakes and solving MWP correctly. Both tasks operate under few-shot settings, with specific prompt details provided in Appendix D.

**Models.** To evaluate LLMs' mathematical reasoning capabilities, we utilize foundational LLMs, SLMs and math-finetuned SLMs.

1. **LLMs:** We utilize 6 LLMs that have shown tremendous performance in MWPs such as GPT-4o, GPT-4, GPT-3.5Turbo, Claude-3-Opus, Llama-2-70b (Touvron et al., 2023), Llama-3-70B (Dubey et al., 2024).
2. **SLMs.** Additionally, we evaluate six popular SLMs—Phi-3-mini (Abdin et al., 2024b), Mixtral-8x7B (Jiang et al., 2024), Llama-2-7b-chat (Touvron et al., 2023), Qwen2-7B (Yang et al., 2024), Llama-3-8B (Dubey et al., 2024), and Llama-3-8b-finetuned (Chen & Li, 2024) trained on high-quality data to assess their reasoning capabilities. Appendix E provides the details of the models, including their last training date.

**Metrics.** We compute the F1 score for all experiments as follows: for mistake detection, the model outputs either "yes" (indicating correct reasoning) or "no" (indicating incorrect reasoning). The ground truth labels are similarly "yes" for correct reasoning and "no" for incorrect reasoning, and the model's predictions are compared against these labels to calculate the F1 score. For performance evaluation, the generated final answer is compared to the ground truth final answer to compute the F1 score for accuracy.

## 4 RESULTS AND ANALYSIS

### 4.1 MISTAKE DETECTION ANALYSIS WITH SIMPLE MWPS

We evaluated the models' ability to detect reasoning mistakes using the SVAMP dataset, which contains simple arithmetic word problems (up to a 4th-grade level) and variations testing question sensitivity, reasoning ability, and structural invariance (Appendix B for more details on the variations included). Mistakes were introduced using both rule-based methods and outputs from SLMs, with human validation ensuring accuracy.

Table 2: Mistake Detection Performance (F1 score) on SVAMP dataset with all variations

| Model | Question Sensitivity | | | Reasoning Ability | | | Structural invariance | | |
|---|---|---|---|---|---|---|---|---|---|
| | Same Object, Different Structure | Different Object, Same Structure | Different Object, Different Structure | Add relevant information | Change Information | Invert Operation | Change order of objects | Change order of phrases | Add irrelevant information |
| GPT-4o | 0.78 | 0.70 | **0.77** | 0.74 | 0.78 | 0.72 | 0.74 | 0.75 | 0.76 |
| GPT-4 | 0.65 | 0.58 | 0.64 | 0.62 | 0.66 | 0.55 | 0.62 | 0.60 | 0.64 |
| GPT-3.5Turbo | 0.76 | 0.77 | 0.71 | 0.75 | 0.77 | 0.74 | 0.78 | 0.72 | 0.75 |
| Llama-2-7b-chat | 0.14 | 0.09 | 0.13 | 0.10 | 0.17 | 0.08 | 0.13 | 0.15 | 0.13 |
| Phi-3-mini | 0.77 | 0.64 | 0.68 | 0.72 | 0.73 | 0.67 | 0.69 | 0.68 | 0.69 |
| Qwen2-7B-Instruct | 0.57 | 0.42 | 0.62 | 0.54 | 0.59 | 0.46 | 0.61 | 0.51 | 0.59 |
| Llama-3-8b | 0.80 | 0.79 | 0.79 | 0.82 | 0.79 | 0.75 | 0.81 | 0.75 | 0.78 |
| Llama-3-70b | 0.73 | 0.69 | 0.73 | 0.70 | 0.73 | 0.69 | 0.70 | 0.70 | 0.73 |
| Llama-3-8b-finetuned | **0.82** | **0.79** | 0.76 | **0.81** | **0.83** | **0.81** | **0.79** | **0.79** | **0.86** |

Table 7 shows presents the models' mistake detection performance across these variations. The results show that none of the models consistently detected mistakes, with F1 scores across all variations falling below 80%. The highest F1 score, 81%, was achieved by Llama-3-8b-finetuned, a fine-tuned model specifically trained on 13 math-related datasets, which outperformed even more advanced models like GPT-4o and GPT-4. This suggests that fine-tuning on domain-specific data offers significant benefits for mathematical tasks.

Despite these improvements, even the fine-tuned model showed significant sensitivity to problem variations. When question sensitivity variations were introduced, performance dropped by 0.08,

Table 3: Mistake Detection Performance (F1 score) on `MWP-MISTAKE` dataset for Task T1. (D-Default reasoning steps, SM-Smaller model reasoning steps) (Bold: Best)

| Model | GSM8K | | MATH | | MATHBENCH | | JEEBENCH | | SVAMP | | AVERAGE | | |
|---|---|---|---|---|---|---|---|---|---|---|---|---|---|
| | D | SM | D | SM | D | SM | D | SM | D | SM | D | SM | Overall |
| **GPT-4o** | **0.85** | **0.86** | **0.83** | **0.94** | **0.80** | **0.99** | **0.79** | **0.99** | 0.74 | 0.92 | **0.80** | **0.94** | **0.87** |
| **GPT-4** | 0.72 | 0.72 | 0.78 | 0.90 | 0.51 | 0.90 | 0.81 | 0.87 | 0.61 | 0.89 | 0.69 | 0.86 | 0.77 |
| **GPT-3.5Turbo** | 0.80 | 0.70 | 0.80 | 0.60 | 0.50 | 0.34 | 0.54 | 0.46 | 0.75 | 0.69 | 0.68 | 0.56 | 0.62 |
| **Claude-3-Opus** | 0.79 | 0.89 | 0.73 | 0.90 | 0.68 | 0.92 | 0.69 | 0.88 | 0.77 | **0.93** | 0.73 | 0.90 | 0.82 |
| **Qwen2-7B** | 0.59 | 0.26 | 0.64 | 0.53 | 0.49 | 0.67 | 0.60 | 0.60 | 0.53 | 0.61 | 0.57 | 0.53 | 0.55 |
| **Phi-3-mini** | 0.70 | NA | 0.65 | NA | 0.54 | NA | 0.55 | NA | 0.70 | NA | 0.63 | NA | 0.63 |
| **Mixtral-8x7B** | 0.73 | NA | 0.79 | NA | 0.62 | NA | 0.70 | NA | 0.64 | NA | 0.70 | NA | 0.70 |
| **Llama-2-7b-chat** | 0.07 | NA | 0.16 | NA | 0.08 | NA | 0.36 | NA | 0.12 | NA | 0.16 | NA | 0.16 |
| **Llama-2-70b** | 0.63 | 0.73 | 0.77 | 0.61 | 0.81 | 0.98 | 0.54 | 0.75 | 0.71 | 0.45 | 0.69 | 0.70 | 0.70 |
| **Llama-3-8B** | 0.79 | 0.81 | 0.79 | 0.79 | 0.56 | 0.58 | 0.50 | 0.67 | 0.78 | 0.81 | 0.68 | 0.73 | 0.71 |
| **Llama-3-8b-finetuned** | 0.83 | 0.82 | 0.83 | 0.76 | 0.77 | 0.80 | 0.55 | 0.74 | **0.81** | 0.68 | 0.76 | 0.76 | 0.76 |
| **Llama-3-70B** | 0.79 | 0.74 | 0.76 | 0.78 | 0.55 | 0.76 | 0.59 | 0.61 | 0.70 | 0.82 | 0.68 | 0.74 | 0.71 |

while reasoning ability and structural invariance variations resulted in reductions of 0.06 and 0.02, respectively. GPT-4o exhibited a similar performance decline, suggesting that even the most advanced models are vulnerable to small variations in problem structure (See Table 7, Appendix B).

These findings highlight a key gap: even on relatively simple problems, models fail to generalize when minor variations are introduced. This suggests that fine-tuning, while beneficial, is insufficient to fully address the deeper issues in mistake detection across mathematical reasoning tasks. More robust methods are needed to improve generalization.

## 4.2 CAN LLMs EFFECTIVELY IDENTIFY MISTAKES IN REASONING STEPS?

We evaluate the ability of various models to detect mistakes in the reasoning steps of MWPs, with F1 scores across five datasets, as shown in Table 3 for both default (D) and smaller models (SM).

**GPT-4o Performance.** GPT-4o is the top performer, outperforming models like GPT-4, GPT-3.5Turbo, and several smaller models. However, despite an overall F1 score of 87% across all datasets, it struggles with consistent mistake detection, particularly on simpler datasets such as SVAMP, and on more complex datasets like JEEBENCH, where its performance notably declines. For instance, on JEEBENCH, GPT-4o 's F1 score drops by 6% compared to its performance on GSM-8K. This shows that while GPT-4o excels in many areas, its precision for comprehensive mistake detection is still lacking, especially when faced with varying levels of complexity.

**Rule-based vs. SLM-generated Mistakes.** One notable observation is that GPT-4o and other models detect SLM-generated mistakes with higher accuracy compared to rule-based mistakes. For instance, GPT-4o achieves an F1 score of 94% for SLM errors versus 80% for rule-based errors across all datasets. This discrepancy suggests potential exposure to SLM-generated data during GPT-4o 's training, giving it an advantage in detecting these mistakes. This is an important insight into how training data might influence the model's effectiveness in mistake detection.

**Performance of GPT-4 vs. GPT-3.5Turbo.** While GPT-3.5Turbo performs similarly to GPT-4 and even surpasses it on certain datasets like GSM-8K, it struggles with errors generated by smaller models. On these, GPT-4 handles mistakes more effectively, likely due to potential data contamination or overfitting during its training. For instance, GPT-4 's F1 score for smaller model-generated mistakes is 86%, compared to GPT-3.5Turbo 's 56%.

**Smaller Models and Fine-tuning.** Smaller fine-tuned models, such as Llama-3-8b-finetuned, demonstrate a competitive performance close to that of GPT-4. Llama-3-8b-finetuned achieves an F1 score of 76%, outperforming other SLMs and even rivaling GPT-4 (77%) in certain cases. This highlights the effectiveness of domain-specific fine-tuning, especially for mathematical tasks, where tailored training significantly improves mistake detection accuracy.

**Challenges with Newer Datasets.** All models, including GPT-4o, face significant challenges with newer and more complex datasets such as MATHBENCH and JEEBENCH. For instance, GPT-4o 's F1 score drops by 6% on JEEBENCH compared to its performance on GSM-8K. This stark decline across models shows that their reasoning capabilities do not generalize well to unseen and more complex problem types. While GPT-4o still leads the pack, its limitations on these datasets underscore the need for better generalization in handling deeper reasoning challenges. Appendix F shows additional detailed results showcasing the F1 score analysis on different types of rule-based reasoning mistakes across different models.

Table 4: Performance in deriving correct answers (F1 score) on `MWP-MISTAKE` dataset for Task T1. (D-Default reasoning steps, SM-Smaller model reasoning steps) (Bold: Best)

| Model | GSM8K | | MATH | | MATHBENCH | | JEEBENCH | | SVAMP | | AVERAGE | | |
|---|---|---|---|---|---|---|---|---|---|---|---|---|---|
| | D | SM | D | SM | D | SM | D | SM | D | SM | D | SM | Overall |
| **GPT-4o** | **0.99** | **0.86** | **0.90** | **0.79** | **0.90** | **0.69** | **0.48** | **0.47** | **1.00** | 0.78 | **0.85** | **0.72** | **0.79** |
| **GPT-4** | 0.97 | 0.77 | 0.80 | 0.65 | 0.88 | 0.46 | 0.20 | 0.27 | 0.98 | 0.73 | 0.77 | 0.57 | 0.67 |
| **GPT-3.5Turbo** | 0.89 | 0.43 | 0.69 | 0.23 | 0.75 | 0.20 | 0.62 | 0.14 | 0.95 | 0.37 | 0.78 | 0.27 | 0.53 |
| **Claude-3-Opus** | 0.98 | 0.86 | 0.89 | 0.90 | 0.92 | 0.51 | 0.46 | 0.32 | 0.98 | **0.80** | 0.84 | 0.68 | 0.76 |
| **Qwen2-7B-Instruct** | 0.92 | 0.50 | 0.81 | 0.35 | 0.85 | 0.28 | 0.72 | 0.16 | 0.90 | 0.30 | 0.84 | 0.32 | 0.58 |
| **Phi-3-mini** | 0.88 | NA | 0.51 | NA | 0.63 | NA | 0.32 | NA | 0.76 | NA | 0.62 | NA | 0.62 |
| **Mixtral-8x7b** | 0.87 | NA | 0.67 | NA | 0.70 | NA | 0.16 | NA | 0.90 | NA | 0.66 | NA | 0.66 |
| **Llama-2-7b-chat** | 0.80 | NA | 0.27 | NA | 0.40 | NA | 0.08 | NA | 0.66 | NA | 0.44 | NA | 0.44 |
| **Llama-2-70b** | 0.67 | 0.20 | 0.43 | 0.08 | 0.56 | 0.08 | 0.31 | 0.04 | 0.78 | 0.06 | 0.55 | 0.09 | 0.32 |
| **Llama-3-8b** | 0.84 | 0.46 | 0.50 | 0.19 | 0.70 | 0.11 | 0.80 | 0.06 | 0.83 | 0.30 | 0.74 | 0.23 | 0.48 |
| **Llama-3-8b-finetuned** | 0.92 | 0.37 | 0.52 | 0.10 | 0.76 | 0.14 | 0.77 | 0.10 | 0.94 | 0.17 | 0.78 | 0.18 | 0.48 |
| **Llama-3-70b** | 0.83 | 0.72 | 0.78 | 0.46 | 0.80 | 0.29 | 0.63 | 0.22 | 0.83 | 0.69 | 0.77 | 0.48 | 0.62 |

### 4.3 CAN LLMS ACCURATELY DERIVE CORRECT ANSWERS DESPITE MISTAKES?

We assess the models' ability to generate correct answers even when reasoning steps contain mistakes. Table 4 presents the F1 scores for Task 1, where models are explicitly tasked with detecting and rectifying mistakes to compute the final answer.

**GPT-4o Performance.** GPT-4o achieves an overall F1 score of 79% across all datasets, demonstrating an impressive ability to derive correct answers despite flawed reasoning. Specifically, it performs exceptionally well on simpler datasets like GSM-8K (99%), MATH (90%), and MATHBENCH (90%) in rectifying rule-based reasoning errors. However, its performance plummets to 48% on the more complex JEEBENCH dataset. Similar trend is seen in mistakes with SLMs, however this performance drop highlights a critical limitation: even though GPT-4o detects SLM based reasoning mistakes with over 90% accuracy, its ability to rectify them and generate correct answers is inconsistent, with F1 scores falling to 70-80%. This suggests that when faced with simple, rule-based mistakes, GPT-4o can often produce the correct answer, either through error correction or data memorization. However, when confronted with more intricate, SLM-generated mistakes, GPT-4o struggles to correct the errors and derive the correct answer, exposing significant shortcomings in the model's reasoning capabilities.

**Performance of Other Models.** Similar trends are observed for other models. Claude-3-Opus and GPT-4 rank second and third, respectively, in terms of performance. SLMs such as Phi, Llama, and Mixtral perform notably worse, with F1 scores ranging between 40-60%, significantly lower than GPT-4o and GPT-4. These results suggest that larger models like GPT-4o have a clear advantage in mistake rectification compared to smaller and fine-tuned models.

**Challenges with Complex Datasets.** All models, including GPT-4o, perform poorly on complex datasets like JEEBENCH, where the ability to derive correct answers drops significantly. This sharp decline underscores a critical limitation of current LLMs: their lack of robustness when confronted with deeper reasoning tasks and more intricate problem sets.

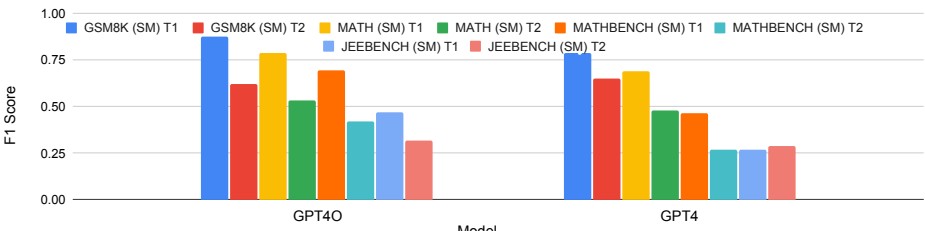

Figure 3: Performance in deriving final answer between T1 and T2. A significant drop in performance when the model does not rectify the incorrect reasoning steps.

**Comparing Performance on Task 2: Identifying Mistakes Without Correction.** In Task 2, models are required to identify the presence of a mistake but are not explicitly tasked with correcting it before providing the final answer. Figure 3 illustrates the F1 scores of GPT-4o, GPT-4, and GPT-3.5Turbo across all datasets for both Task 1 (detect and rectify mistakes) and Task 2 (identify mistakes and compute the answer without rectification).

**GPT-4o Performance.** There is a noticeable drop in GPT-4o's performance between Task 1 and Task 2 across all datasets. In Task 1, where the model is prompted to both detect and correct mistakes, GPT-4o achieves higher accuracy, particularly on simpler datasets. However, in Task 2, where it only identifies whether a mistake is present, its F1 score significantly decreases. This decline suggests that

GPT-4o lacks the inherent ability to rectify mistakes unless it is explicitly instructed to do so. This inability to naturally correct mistakes without guidance reveals a key weakness in its reasoning.

**GPT-4 Performance.** While GPT-4 follows a similar trend to GPT-4o in showing a performance drop from Task 1 to Task 2, the gap between its Task 1 and Task 2 performance is smaller. This indicates that although GPT-4's overall performance is lower than GPT-4o, it experiences less of a drop when transitioning between the two tasks. This could suggest that GPT-4 is more consistent in detecting mistakes but, like GPT-4o, struggles to correct them when not explicitly prompted. The lower overall performance compared to GPT-4o indicates that GPT-4 is less capable of achieving high accuracy on both tasks.

### 4.4 Exploring Data Contamination and Memorization Effects in Math Reasoning Tasks

In our analysis of LLMs' mathematical reasoning performance, we've identified potential instances of data contamination and memorization, both of which can significantly impact the effectiveness of these models. Data contamination, characterized by the presence of test data from downstream tasks in LLMs' training data, poses a major challenge in accurately assessing their real-world performance. Meanwhile, memorization occurs when models replicate solutions from training data without grasping the underlying principles, thereby hindering their ability to generalize to new problems.

The presence of data contamination is evident in instances of unexpectedly high performance on certain datasets. For example, GPT-3.5Turbo's superior performance over GPT-4 on the GSM-8K dataset raises concerns about biases in GPT-4's training data. Similarly, the comparable performance between smaller and larger models suggests the potential presence of memorization. These findings underscore the critical need for rigorous evaluation to mitigate the impacts of memorization, ensuring the reliability and effectiveness of LLMs in real-world applications.

Investigating data contamination and memorization poses challenges due to restricted pre-training data access and computational limitations. To tackle this, we employ an approach outlined in (Golchin & Surdeanu, 2024), utilizing an LLM to replicate individual instances of the dataset. This involves guiding the LLM with instructions containing unique identifiers from the source dataset, like dataset name, partition (e.g., train, test, or validation), and a fragment of the reference instance. By instructing the LLM to complete these partial instances, we can evaluate contamination and memorization.

To detect contamination, a heuristic is applied comparing the average overlap score between generated completions and reference instances using ROUGE-L (Lin, 2004). This comparison is made between guided instructions (including dataset and partition identifiers) and general instructions (lacking such identifiers). If the overlap score is significantly larger with guided instructions, it suggests contamination. This method relies on the premise that the only distinction between the two instructions is the inclusion of dataset and partition names in guided instructions, implying any improvement can be attributed to contamination (Appendix I for more details).

Figure 4 illustrates the difference in ROUGE-L scores between guided and general instructions across all datasets for various models. The results highlight notable discrepancies, providing early evidence of data contamination, particularly among the larger models.

**GPT-4o Performance.** GPT-4o exhibits the highest ROUGE-L scores across all datasets, suggesting a significant level of data contamination. This is consistent with its earlier performance, where it excelled in simpler tasks but struggled with more complex datasets, likely due to reliance on memorized data rather than true reasoning capabilities.

**Comparative Contamination Across Models.** Following GPT-4o, both GPT-4 and GPT-3.5Turbo show progressively lower ROUGE-L scores, though they still indicate some level of contamination. This pattern reinforces the earlier performance trends, where these models performed well but not as dominantly as GPT-4o, suggesting that their performance may also benefit from memorized data (especially on GSM-8K), albeit to a lesser degree.

**SLMs' Minimal Contamination.** In contrast, smaller language models (SLMs) such as Llama and Phi display negative ROUGE-L scores, suggesting minimal to no contamination. These models seem to rely more on reasoning rather than memorization, as their performance is not inflated by exposure to the test data during training. However, their lower overall performance on complex tasks highlights that they lack the advanced reasoning capabilities needed to match the larger models.

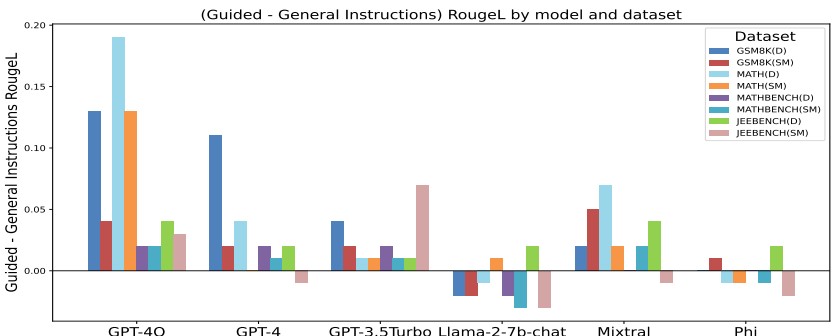

Figure 4: Difference between guided and general instructions rouge-L score across all datasets. A high positive difference indicates high contamination and a low positive or negative difference indicates, little to no contamination.

Table 5: Ability to Rectify mistakes and derive correct final answer on `MWP-MISTAKE` dataset for Task T1. (D-Default reasoning steps, SM-Smaller model reasoning steps) (Bold: Best)

| Model | GSM8K | | MATH | | MATHBENCH | | JEEBENCH | | SVAMP | | AVERAGE | | |
|---|---|---|---|---|---|---|---|---|---|---|---|---|---|
| | D | SM | D | SM | D | SM | D | SM | D | SM | D | SM | Overall |
| **GPT-4o** | **0.98** | **0.91** | **0.87** | **0.84** | **0.90** | **0.64** | 0.42 | **0.42** | **1.00** | **0.86** | **0.83** | **0.73** | **0.78** |
| **GPT-4** | 0.96 | 0.88 | 0.72 | 0.70 | 0.83 | 0.45 | 0.10 | 0.24 | 0.94 | 0.77 | 0.71 | 0.61 | 0.66 |
| **GPT-3.5Turbo** | 0.81 | 0.56 | 0.54 | 0.34 | 0.62 | 0.34 | 0.16 | 0.05 | 0.93 | 0.57 | 0.61 | 0.37 | 0.49 |
| **Claude-3-Opus** | 0.97 | 0.94 | 0.84 | 0.89 | 0.87 | 0.57 | 0.27 | 0.33 | 0.96 | 0.85 | 0.78 | 0.72 | 0.75 |
| **Qwen2-7B-Instruct** | 0.83 | 0.51 | 0.77 | 0.47 | 0.69 | 0.29 | 0.29 | 0.21 | 0.78 | 0.50 | 0.67 | 0.40 | 0.53 |
| **Phi-3-mini** | 0.79 | NA | 0.37 | NA | 0.41 | NA | 0.03 | NA | 0.63 | NA | 0.45 | NA | 0.45 |
| **Mixtral-8x7b** | 0.77 | NA | 0.56 | NA | 0.57 | NA | 0.17 | NA | 0.83 | NA | 0.58 | NA | 0.58 |
| **Llama-2-7b-chat** | 0.73 | NA | 0.21 | NA | 0.11 | NA | 0.04 | NA | 0.52 | NA | 0.32 | NA | 0.32 |
| **Llama-2-70b** | 0.57 | 0.25 | 0.34 | 0.07 | 0.46 | 0.06 | 0.02 | 0.03 | 0.60 | 0.21 | 0.40 | 0.12 | 0.26 |
| **Llama-3-8b** | 0.77 | 0.51 | 0.39 | 0.24 | 0.58 | 0.08 | **0.49** | 0.06 | 0.65 | 0.39 | 0.58 | 0.26 | 0.42 |
| **Llama-3-8b-finetuned** | 0.85 | 0.41 | 0.33 | 0.10 | 0.69 | 0.18 | 0.25 | 0.13 | 0.91 | 0.26 | 0.60 | 0.21 | 0.41 |
| **Llama-3-70b** | 0.80 | 0.88 | 0.72 | 0.62 | 0.73 | 0.33 | 0.21 | 0.21 | 0.83 | 0.81 | 0.66 | 0.57 | 0.61 |

## 4.5 CAN LLMs CORRECTLY RECTIFY MISTAKES IN REASONING STEPS?

In Task 1, the model not only detects mistakes but also attempts to rectify them to arrive at the correct answer. We evaluate the model's ability to rectify mistakes once detected by examining the number of questions where mistakes were identified and calculating how many times the model produced the correct answer after rectification. The assumption is that if the model reaches the correct final answer after detecting a mistake, it has successfully rectified the incorrect reasoning step. For instance, if the model identifies mistakes in 90 out of 100 questions and rectifies them in 45 cases (resulting in final correct answer), the rectification score would be 50% (45/90).

Table 5 illustrates the performance of different models in rectifying reasoning steps and producing the correct final answer across various datasets.

GPT-4o shows high proficiency in rectifying mistakes, achieving an overall rectification score of 78% across all datasets. It outperforms GPT-4 by 11% and exceeds other models, including SLMs, by over 35%. Specifically, GPT-4o excels in correcting mistakes caused by rule-based reasoning compared to those induced by SLMs. However, its ability to fix mistakes decreases with more complex datasets like MATHBENCH and JEEBENCH. On simpler datasets, such as GSM-8K, MATH, and SVAMP, GPT-4o demonstrates high accuracy in rectification, potentially due to either data contamination (as discussed earlier) or the simpler nature of rule-based mistakes.

As observed earlier, Claude-3-Opus performs comparably to GPT-4o in rectifying mistakes. Other models, however, exhibit poorer rectification abilities, with scores ranging between 30-50%. Notably, Llama-3-70B achieves performance similar to GPT-4, indicating strong rectification capabilities.

To delve deeper into the rectification process, we also compute the percentage of questions where the model rectified the reasoning steps but still arrived at incorrect answers. Across the `MWP-MISTAKE` dataset, GPT-4o failed to derive the correct answer in 17% of cases after correcting the reasoning, while other models like GPT-4, GPT-4, Llama-2-7b-chat, Mixtral-8x7B, and Phi-3-mini resulted in 30%, 43.5%, 80.9%, 40.2%, and 55.6% incorrect answers, respectively. Additionally, we evaluated the rectified reasoning steps by comparing them with ground-truth reasoning steps to assess the effectiveness and alignment of the rectification process across models (detailed in Appendix G and H). This comparison is quantified using traditional NLP metrics such as BERTScore.

## 4.6 How does OpenAI o1 model perform on MWP-MISTAKE dataset?

OpenAI recently introduced the O1 model, designed to enhance reasoning capabilities by spending more time processing tasks before responding. In a preliminary analysis comparing the performance of O1 and GPT-4O on 120 questions from the complex JEEBENCH dataset (38 correct reasoning steps, 56 rule-based mistakes, and 26 SLM-generated mistakes), several key insights emerged. O1 consistently outperforms GPT-4O, particularly in deriving correct final answers, showcasing its superior reasoning abilities across complex tasks (Table 15 Appendix J).

**Rule-based mistake identification.** both O1 and GPT-4O perform similarly, with F1 scores of 0.4759 and 0.45, respectively. This suggests that both models struggle to consistently identify simple rule-based errors, detecting them with less than 50% accuracy. However, the significant divergence becomes apparent when comparing their ability to derive the correct final answer despite the mistakes. While GPT-4O manages an F1 score of only 0.43, O1 excels with a final answer F1 score of 0.8277, showing a notable improvement in reasoning capabilities. O1's ability to achieve such high final accuracy, despite similar mistake detection rates, underscores its advanced reasoning abilities, which may benefit from improved rectification strategies or more sophisticated handling of mistakes during the reasoning process.

**SLM-generated mistakes.** When analyzing SLM-generated mistakes, both models achieve 100% mistake detection accuracy, reflecting strong capabilities in identifying these more complex errors. However, the models diverge significantly in their ability to correct these mistakes and derive the correct final answer. O1 reaches a final answer F1 score of 0.9, while GPT-4O lags significantly behind with a score of only 0.62. This stark contrast highlights O1's substantial advancement not only in detecting mistakes but also in rectifying them to produce the correct final answer, showcasing its enhanced reasoning and generalization capabilities on more challenging datasets.

In summary, while both models are comparable in terms of mistake identification, O1 demonstrates a clear advantage in final answer generation and rectification, particularly on SLM-based mistakes. These results illustrate the superior reasoning capabilities of O1 over GPT-4O, making it a more effective model for handling intricate reasoning tasks. However, issues like potential data contamination and inefficiencies in processing time and token usage with O1 remain areas for further optimization.

## 5 Key Insights, Takeaways, and Potential Directions for Improving Mathematical Reasoning

1. **GPT-4o's Performance Strengths and Limitations:** GPT-4o is the top performer across all datasets, achieving an overall F1 score of 87%. Its foundational capabilities enable strong performance in both mistake detection and rectification, particularly on simpler datasets like GSM-8K. However, its performance drops significantly on more complex datasets such as JEEBENCH and MATHBENCH, indicating limitations in handling highly complex or novel problems and highlighting the need for improved reasoning capabilities.
2. **Data Contamination and Overfitting Concerns:** GPT-4o's unexpectedly high performance on datasets like GSM-8K and SVAMP suggests possible data contamination and overfitting, with models benefiting from memorized examples. To ensure fair evaluation, cleaner datasets and rigorous training methods are needed to assess true reasoning abilities rather than memorization.
3. **Challenges with Smaller Language Models (SLMs):** There is a notable performance gap between smaller models (SLMs) and larger models like GPT-4o. While some SLMs, like Llama-3-8b-finetuned and Llama-3-70B, show competitive results, this may stem from similar contamination issues. Enhancing SLMs' reasoning abilities, is a key area for improvement.
4. **Generalization Difficulties Across Datasets:** The performance decline on newer datasets such as MATHBENCH and JEEBENCH points to a generalization issue in LLMs. While models perform well on familiar datasets, they struggle with novel problems. Addressing this requires improved training strategies and more diverse datasets to broaden models' reasoning skills.
5. **Inconsistent Rectification Abilities:** Despite strong mistake detection, GPT-4o shows inconsistent rectification performance, especially in complex datasets. Its ability to correct errors drops significantly between simple tasks (like rule-based errors) and more challenging ones (SLM-generated mistakes). This highlights the need for more robust error correction capabilities in diverse reasoning scenarios.

Future research should prioritize cleaner datasets and techniques to reduce data contamination and overfitting, enabling better generalization to new tasks. Improving error rectification and enhancing smaller models through fine-tuning are key, as is advancing models' ability to handle complex rule-based reasoning for better performance on structured problems.

# 6 RELATED WORK

Current research on large language models (LLMs) for solving math word problems (MWPs) primarily emphasizes generating correct answers, often focusing on overall accuracy rather than evaluating the underlying reasoning processes. Studies like MathPrompter (Imani et al., 2023) and WizardMath (Luo et al., 2023) showcase impressive results in solving MWPs by generating complex reasoning steps. However, their focus remains heavily centered on achieving the correct answer without rigorously evaluating the correctness, relevance, or verification of the individual reasoning steps. For instance, works such as (Liu et al., 2024b; Yuan et al., 2023; Schulman et al., 2017) focus primarily on enhancing LLMs' ability to reach accurate answers but do not delve into assessing whether the reasoning process itself is correct or aligned with logical problem-solving paths.

Several recent works have begun shifting their attention toward reasoning quality, but these efforts remain limited in scope. Studies like (Sawada et al., 2023) evaluate reasoning by comparing the similarity of generated and reference reasoning, while others, such as (Xia et al., 2024), introduce the idea of assessing reasoning steps through metrics like validity and redundancy. ROSCOE(Golovneva et al., 2023) takes this further by offering a suite of unsupervised metrics that evaluate various aspects of reasoning quality, such as semantic consistency and logicality, rather than just the final answer. While these methods attempt to scrutinize reasoning steps, they often fall short of addressing the detection and rectification of specific reasoning mistakes within MWPs, leaving a gap in understanding how well LLMs can manage flawed reasoning.

A third significant gap in the literature pertains to the limited exploration of LLMs' foundational reasoning abilities, particularly in mistake detection and rectification. While some works propose LLMs as verifiers for their own reasoning (Zhang et al., 2024b; Zheng et al., 2023), they typically assess reasoning correctness without tackling the deeper issue of identifying and correcting logical mistakes. Moreover, studies like (Olausson et al., 2024) demonstrate that LLMs struggle to find and correct their own reasoning errors, especially in tasks involving code generation. Recent works such as Alice in Wonderland (Nezhurina et al., 2024) breakdown the function and reasoning capabilities of LLMs and show that even small variations in such common sense tasks has drastic performance reduction. However, there remains a lack of rigorous benchmarking for mistake detection and correction in MWPs, especially for foundational models.

Our work addresses this gap by introducing the MWP-Mistake dataset, which includes diverse, systematic reasoning mistakes in MWPs. Unlike prior research, our analysis focuses not only on models' ability to detect mistakes but also on their ability to rectify these errors and generate correct answers. We provide a comprehensive benchmark, comparing state-of-the-art models on both simple and complex datasets. Through this work, we aim to provide a clearer understanding of current models' limitations in handling reasoning mistakes and propose a framework for evaluating LLMs' true reasoning abilities, rather than relying on answer accuracy alone.

# 7 CONCLUSIONS

This study evaluates large language models (LLMs) such as GPT-4o, GPT-4, GPT-3.5Turbo, along-side smaller models like Llama-2-7b-chat, Mixtral-8x7B, and Phi-3-mini, on their ability to detect and correct errors in mathematical reasoning. Using our `MWP-MISTAKE` dataset, which includes incorrect reasoning steps generated through both rule-based methods and smaller models, we comprehensively assess LLMs' performance in error detection and rectification. While GPT-4o outperforms other models, there remains a gap in its ability to consistently detect mistakes, as it struggles with several simple problems and its performance degrades on more complex tasks. We also uncover issues of data contamination and overfitting, especially in GPT-4's performance on GSM8K, and observe a performance drop on newer datasets like MATHBENCH and JEEBENCH, highlighting generalization challenges. Addressing these limitations—such as enhancing generalization and minimizing data contamination—is essential for making LLMs more reliable and applicable to real-world mathematical problem-solving. Future research should focus on refining training processes and strengthening models' reasoning abilities to meet these challenges.

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

## APPENDIX

The dataset and code to run all experiments are provided in this repository.

## A   MWP-MISTAKE DATASET

MWP-MISTAKE dataset is curated using 4 different types of well-known datasets. Below are the details of each of the datasets.

- SVAMP Patel et al. (2021): SVAMP is a MWP dataset created by applying a carefully chosen variations over examples sampled from existing datasets, AsDiv-A Miao et al. (2020) and MAWPS Koncel-Kedziorski et al. (2016).
- GSM-8K Cobbe et al. (2021b):GSM-8K is a dataset of diverse grade school math word problems created by human writers, involving basic arithmetic operations. Released in November 2021.

- MATH Hendrycks et al. (2021): The MATH dataset is divided into seven categories, each with five difficulty levels. For our study, we used levels 1, 2, and 3 from the algebra and counting and probability categories. Released in November 2021. We focused on Levels 1 to 3 because the problems in Levels 4 and 5 are more complex, requiring specific notations, symbols, and equations. Injecting reasoning mistakes into such complex problems is non-trivial and would require expert knowledge to ensure accuracy in the reasoning chains.
- MATHBENCH Liu et al. (2024a): MATHBENCH is a recent dataset with questions divided by educational stages, from basic arithmetic to college levels. For our experiment, we chose middle and high-school-level single-choice multiple-choice questions. Released in May 2024.
- JEEBENCH Arora et al. (2023): JEEBENCH is a challenging benchmark dataset for evaluating LLM problem-solving abilities, containing 515 pre-engineering math, physics, and chemistry problems from the IIT JEE-Advanced Exam. For our experiment, we chose mathematics single-choice questions only. Released in October 2023.

## A.1 Prompts to curate reasoning steps in `MWP-MISTAKE` dataset

GSM-8K and MATH already contain MWP questions, a chain of thought reasoning steps and a final answer. To curate chain of thought reasoning step for MATHBENCH and JEEBENCH we made use of GPT-4. While prompting GPT-4 we made sure that reasoning steps did not contain the final answer, so that final answer is not picked directly from the reasoning step. Listing 1 prompt is used to curate the reasoning steps.

```
1 Strictly follow the below conditions.
2 1. Output format: \nReasoning Chain: \nFinal Answer:
3 2. Reasoning Chain should be separated by a new line only.
4 3. Reasoning chain cannot have the final answer. (Replace the
     final answer  in the reasoning chain with its calculation or
     ####)
5 4. Do not include any additional information in the final answer (
     only the answer).
```

Listing 1: Prompt to curate reasoning chain without answers.

Table 6 shows examples of default reasoning steps from GSM-8K dataset.

# B SVAMP Variations

Figure 5 shows the 9 different types of carefully curated variations sampled from existing datasets, AsDiv-A Miao et al. (2020) and MAWPS Koncel-Kedziorski et al. (2016). Across each category, we evaluated the mistake detection performance of each model Table 7 shows that Llama-3-8b-finetuned performed the best due to this preexisting knowledge on solving MWPs, achiving the highest average F1 score of 81% across the variations. We also evaluated model's sensitivity to variations, table 8 shows the max performance change across different models. GPT-4o performance significantly dropped by 0.08, 0.06, and 0.02 when exposed to variations related to question sensitivity, reasoning ability, and structural invariance, respectively. Examples of variations are as follows:

- Reasoning ability - Change Information: This involves changing entities, e.g., replacing "Jack" with "Dorothy."
- Reasoning ability - Invert Operation: Here, operations or calculations are altered while keeping the rest of the structure the same.
- Structural Invariance - Change Order of Objects: This variation reverses the order of entities, such as changing "8 marbles and 3 stones" to "3 stones and 8 marbles.

Across all these simple variations, we observed a performance drop of 10% in GPT-4 and around 6% in GPT-4o, highlighting the sensitivity of these models. Interestingly, fine-tuned models like Llama-3-8B-Finetuned demonstrated greater robustness, with just a 2% performance drop.

Table 6: Example of rule based incorrect reasoning step (GSM-8K dataset)

| | |
|---|---|
| Question | Gerald spends $100 a month on baseball supplies. His season is 4 months long. He wants to use the months he's not playing baseball to save up by raking, shoveling, and mowing lawns. He charges $10 for each. How many chores does he need to average a month to save up for his supplies? |
| Final Answer | 5 |
| Gold Reasoning step | He needs to save up $400 because 4 x 100 = 400
He has 8 months to earn this money because 12 - 4 = 8
He needs to earn $50 a month because 400 / 8 = 50
He needs to do 5 tasks a month because 50 / 10 = 5 |
| Shuffle reasoning step | He needs to earn $50 a month because 400 / 8 = 50
He needs to save up $400 because 4 x 100 = 400
He needs to do 5 tasks a month because 50 / 10 = 5
He has 8 months to earn this money because 12 - 4 = 8 |
| Delete reasoning step | He needs to save up $400 because 4 x 100 = 400
He needs to earn $50 a month because 400 / 8 = 50
He needs to do 5 tasks a month because 50 / 10 = 5 |
| Shuffle numerical values | He needs to save up $400 because 4 x 100 = 400
He has 50 months to earn this money because 8 - 8 = 4
He needs to earn $12 a month because 400 / 8 = 50
He needs to do 5 tasks a month because 50 / 10 = 5 |
| Replace numerical values | He needs to save up $400 because 4 x 100 = 400
He has 8 months to earn this money because 12 - 4 = 8
He needs to earn $6 a month because 32 / 8 = 50
He needs to do 76 tasks a month because 50 / 10 = 5 |
| Shuffle Operations | He needs to save up $400 because 4 x 100 = 400
He has 8 months to earn this money because 12 * 4 = 8
He needs to earn $50 a month because 400 - 8 = 50
He needs to do 5 tasks a month because 50 / 10 = 5 |
| Insert Random Reasoning step | He needs to save up $400 because 4 x 100 = 400
Therefore, Faye has $60 - $30 = $30 left.
He has 8 months to earn this money because 12 - 4 = 8
He needs to earn $50 a month because 400 / 8 = 50
He needs to do 5 tasks a month because 50 / 10 = 5 |

Table 7: Mistake Detection Performance (F1 score) on SVAMP dataset with all variations)

| Model | Question Sensitivity | | | Reasoning Ability | | | Structural invariance | | |
|---|---|---|---|---|---|---|---|---|---|
| | Same Object, Different Structure | Different Object, Same Structure | Different Object, Different Structure | Add relevant information | Change Information | Invert Operation | Change order of objects | Change order of phrases | Add irrelevant information |
| GPT-4o | 0.78 | 0.70 | **0.77** | 0.74 | 0.78 | 0.72 | 0.74 | 0.75 | 0.76 |
| GPT-4 | 0.65 | 0.58 | 0.64 | 0.62 | 0.66 | 0.55 | 0.62 | 0.60 | 0.64 |
| GPT-3.5Turbo | 0.76 | 0.77 | 0.71 | 0.75 | 0.77 | 0.74 | 0.78 | 0.72 | 0.75 |
| Llama-2-7b-chat | 0.14 | 0.09 | 0.13 | 0.10 | 0.17 | 0.08 | 0.13 | 0.15 | 0.13 |
| Phi-3-mini | 0.77 | 0.64 | 0.68 | 0.72 | 0.73 | 0.67 | 0.69 | 0.68 | 0.69 |
| Qwen2-7B-Instruct | 0.57 | 0.42 | 0.62 | 0.54 | 0.59 | 0.46 | 0.61 | 0.51 | 0.59 |
| Llama-3-8b | 0.80 | 0.79 | 0.79 | 0.82 | 0.79 | 0.75 | 0.81 | 0.75 | 0.78 |
| Llama-3-70b | 0.73 | 0.69 | 0.73 | 0.70 | 0.73 | 0.69 | 0.70 | 0.70 | 0.73 |
| Llama-3-8b-finetuned | **0.82** | **0.79** | 0.76 | **0.81** | **0.83** | **0.81** | **0.79** | **0.79** | **0.86** |

Table 8: Max Performance change with introduction of variations on SVAMP dataset.)

| Model | Question Sensitivity | Reasoning Ability | Structural invariance |
|---|---|---|---|
| **GPT-4o** | 0.08 | 0.06 | 0.02 |
| **GPT-4** | 0.07 | 0.04 | 0.04 |
| **GPT-3.5Turbo** | 0.06 | 0.03 | 0.06 |
| **Llama-2-7b-chat** | 0.05 | 0.09 | 0.02 |
| **Phi-3-mini** | 0.09 | 0.06 | 0.01 |
| **Qwen2-7B-Instruct** | 0.20 | 0.13 | 0.1 |
| **Llama-3-8b** | 0.01 | 0.07 | 0.06 |
| **Llama-3-70b** | 0.04 | 0.04 | 0.03 |
| **Llama-3-8b-finetuned** | 0.06 | 0.02 | 0.07 |

| CATEGORY | VARIATION | EXAMPLES |
|---|---|---|
| Question Sensitivity | Same Object, Different Structure | **Original:** Allan brought two balloons and Jake brought four balloons to the park. How many balloons did Allan and Jake have in the park?
**Variation:** Allan brought two balloons and Jake brought four balloons to the park. How many more balloons did Jake have than Allan in the park? |
| | Different Object, Same Structure | **Original:** In a school, there are 542 girls and 387 boys. 290 more boys joined the school. How many pupils are in the school?
**Variation:** In a school, there are 542 girls and 387 boys. 290 more boys joined the school. How many boys are in the school? |
| | Different Object, Different Structure | **Original:** He then went to see the oranges being harvested. He found out that they harvest 83 sacks per day and that each sack contains 12 oranges. How many sacks of oranges will they have after 6 days of harvest?
**Variation:** He then went to see the oranges being harvested. He found out that they harvest 83 sacks per day and that each sack contains 12 oranges. How many oranges do they harvest per day? |
| Reasoning Ability | Add relevant information | **Original:** Every day, Ryan spends 4 hours on learning English and 3 hours on learning Chinese. How many hours does he spend on learning English and Chinese in all?
**Variation:** Every day, Ryan spends 4 hours on learning English and 3 hours on learning Chinese. If he learns for 3 days, how many hours does he spend on learning English and Chinese in all? |
| | Change Information | **Original:** Jack had 142 pencils. Jack gave 31 pencils to Dorothy. How many pencils does Jack have now?
**Variation:** Dorothy had 142 pencils. Jack gave 31 pencils to Dorothy. How many pencils does Dorothy have now? |
| | Invert Operation | **Original:** He also made some juice from fresh oranges. If he used 2 oranges per glass of juice and he made 6 glasses of juice, how many oranges did he use?
**Variation:** He also made some juice from fresh oranges. If he used 2 oranges per glass of juice and he used up 12 oranges, how many glasses of juice did he make? |
| Structural Invariance | Change order of objects | **Original:** John has 8 marbles and 3 stones. How many more marbles than stones does he have?
**Variation:** John has 3 stones and 8 marbles. How many more marbles than stones does he have? |
| | Change order of phrases | **Original:** Matthew had 27 crackers. If Matthew gave equal numbers of crackers to his 9 friends, how many crackers did each person eat?
**Variation:** Matthew gave equal numbers of crackers to his 9 friends. If Matthew had a total of 27 crackers initially, how many crackers did each person eat? |
| | Add irrelevant information | **Original:** Jack had 142 pencils. Jack gave 31 pencils to Dorothy. How many pencils does Jack have now?
**Variation:** Jack had 142 pencils. Dorothy had 50 pencils. Jack gave 31 pencils to Dorothy. How many pencils does Jack have now? |

Figure 5: Variations in SVAMP dataset on simple Math Problems Patel et al. (2021)

## C  SLMs REASONING STEPS

SLMs were used to generate chain of thought (COT) reasoning step and final answers for all dataset questions. Each model Llama-2-7b-chat, Mixtral-8x7B, Phi-3-mini where prompted using Listing 1 to curate the reasoning step without an answer. If the final answer was incorrect we filtered out the reasoning steps as incorrect. Table 9 shows examples of SLM incorrect reasoning steps from GSM-8K dataset.

Table 9: Example of incorrect reasoning steps generated using SLM's (GSM-8K dataset)

| | |
|---|---|
| Question | Gerald spends $100 a month on baseball supplies.
His season is 4 months long.
He wants to use the months he's not playing baseball to save up by raking, shoveling, and mowing lawns.
He charges $10 for each. How many chores does he need to average a month to save up for his supplies? |
| Final Answer | 5 |
| Gold Reasoning step | He needs to save up $400 because 4 x 100 = 400.
He has 8 months to earn this money because 12 - 4 = 8
He needs to earn $50 a month because 400 / 8 = 50
He needs to do 5 tasks a month because 50 / 10 = 5
#### 5 |
| Llama-2-7b-chat Incorrect reasoning step | Gerald spends $100 on baseball supplies per month.
Season is 4 months long.
To save up for baseball supplies, Gerald needs to earn more than the cost of supplies per month.
Gerald charges $10 for each chore he does.
To calculate the number of chores Gerald needs to do per month,
divide the cost of supplies by the amount he charges per chore. |
| Llama-2-7b-chat Predicted Answer | "" |
| Mixtral-8x7B Incorrect reasoning step | Gerald spends $100 a month for 4 months, so his total spending for the season is 100*4 = $400
Now, we need to find out how many chores he needs to do to earn this amount
He charges $10 per chore, so we divide the total amount by the cost per chore: 400 / 10 = #### |
| Mixtral-8x7B Predicted Answer | 40 |
| Phi-3-mini Incorrect reasoning step | Gerald's total expenditure for baseball supplies = $100 per month * 4 months = $400
Gerald earns $10 per chore.
To find out how many chores he needs to do to save up $400,
we divide his total expenditure by the amount he earns per chore.= $400 / $10= #### |
| Phi-3-mini Predicted Answer | 40 |

## D  TASK T1 AND T2

Task T1 evaluates the model's ability to detect mistakes rectify them and derive the correct answer. Listing 2 was used in a few shot settings for task T1.

```
1 You are a mathematics educator with a deep understanding of
     elementary and middle school mathematics. You are experienced
     in teaching multi-step problem-solving techniques and have a
     knack for breaking down complex problems into manageable steps
     . Your expertise lies in basic arithmetic operations such as
     addition, subtraction, multiplication, and division. You can
     provide clear, step-by-step solutions to mathematical problems
      that require multi-step reasoning.
2
3 You are provided with a mathematical question and a step-by-step
     solution along with it. The solution might have some mistakes.
      Identify if the solution is correct or incorrect. If the
     solution is correct, output the final answer with the help of
     the solution provided. If the solution is incorrect, correct
     the existing solution and determine the final answer with the
     help of the corrected solution.
4 Reasoning chain Correct (Yes/No):
5 Corrected reasoning chain or NA:
6 Final answer (just the number):
```

Listing 2: Prompt for Task T1

Task T2 evaulates the model's ability to detect mistake and solve MWP based on the provided reasoning step. Listing 3 was used in a few shot setting for task T2. Here we insure that final answer is generated with the help of the reasoning steps provided, which may or may not be correct.

```
1 You are a mathematics educator with a deep understanding of
     elementary and middle school mathematics. You are experienced
     in teaching multi-step problem-solving techniques and have a
     knack for breaking down complex problems into manageable steps
     . Your expertise lies in basic arithmetic operations such as
     addition, subtraction, multiplication, and division. You can
     provide clear, step-by-step solutions to mathematical problems
      that require multi-step reasoning.
2
3 You are provided with a mathematical question and a step-by-step
     solution along with it. The solution might have some mistakes.
      Identify if the solution is correct or incorrect and output
     the final answer based on the provided solution.
4 Reasoning chain Correct (Yes/No):
5 Final answer (just the number):
```

Listing 3: Prompt for Task T2

## E  MODEL USED

Below are brief details of the models we have used for benchmarking our MWP-MISTAKE dataset.

1. **GPT-4o:** GPT-4o is a multimodal model by OpenAI, and it has the same high intelligence as GPT-4 Turbo but is much more efficient—it generates text 2x faster and is 50% cheaper. Additionally, GPT-4o has the best vision and performance across non-English languages of any OpenAI model. Last training data: October 2023.

2. **GPT-4:** GPT-4 is a large multimodal model by OpenAI that can solve difficult problems with greater accuracy than any of OpenAI previous models, thanks to its broader general knowledge and advanced reasoning capabilities. Last training data: September 2021.

Table 10: F1 Score Analysis on Different Types of Rule-Based Reasoning Mistakes on GSM8k Dataset.

| Model | Correct Reasoning | Shuffle Reasoning | Delete Reasoning | Shuffle Numerical | Replace Numerical | Shuffle Operations | Random Reasoning | SLM Combined |
|---|---|---|---|---|---|---|---|---|
| GPT-4o | 0.69 | 0.84 | 0.87 | 0.92 | 0.96 | 0.93 | 0.67 | 0.73 |
| GPT-4 | 0.95 | 0.38 | 0.54 | 0.85 | 0.89 | 0.72 | 0.33 | 0.52 |
| GPT-3.5Turbo | 0.83 | 0.65 | 0.71 | 0.82 | 0.87 | 0.78 | 0.76 | 0.52 |
| Llama-2-7b-chat | 1.00 | 0.00 | 0.01 | 0.00 | 0.02 | 0.00 | 0.09 | NA |
| Mixtral-8x7b | 0.83 | 0.59 | 0.60 | 0.77 | 0.75 | 0.63 | 0.63 | NA |
| Phi-3-mini | 0.85 | 0.72 | 0.42 | 0.60 | 0.52 | 0.58 | 0.82 | NA |
| Claude-3-Opus | 0.94 | 0.51 | 0.71 | 0.84 | 0.94 | 0.79 | 0.54 | 0.76 |
| Qwen2-7B-Instruct | 0.95 | 0.45 | 0.36 | 0.53 | 0.45 | 0.34 | 0.62 | 0.13 |
| Llama-2-70b | 0.85 | 0.55 | 0.49 | 0.45 | 0.44 | 0.41 | 0.78 | 0.55 |
| Llama-3-8b | 0.77 | 0.76 | 0.62 | 0.68 | 0.82 | 0.63 | 0.98 | 0.68 |
| Llama-3-70b | 0.75 | 0.52 | 0.60 | 0.83 | 0.93 | 0.87 | 0.84 | 0.54 |
| Llama-3-8b-finetuned | 0.77 | 0.91 | 0.80 | 0.77 | 0.75 | 0.65 | 0.99 | 0.68 |

3. **GPT-3.5Turbo:** GPT-3.5Turbo is a large language model by OpenAI GPT-3.5 that can understand and generate natural language or code and has been optimized for chat using the Chat Completions API but work well for non-chat tasks as well. Last training date: September 2021.

4. **Claude-3-Opus:** Claude-3-Opus is Anthropic's most capable and intelligent model yet, ideal for navigating complex tasks like in-depth analysis, research, and task automation. Last training data: August 2023.

5. **Llama-2-7b-chat:** Llama 2 is a collection of pretrained and fine-tuned generative text models ranging in scale from 7 billion to 70 billion parameters from meta. This is the 7B fine-tuned model, optimized for dialogue use cases. Training date: September 2022.

6. **Mixtral-8x7B:** Mixtral is a Mixture of Experts (MoE) model with 8 experts per MLP, with a total of 45 billion parameters. Despite the model having 45 billion parameters, the compute required for a single forward pass is the same as that of a 14 billion parameter model. This is because even though each of the experts have to be loaded in RAM (70B like ram requirement) each token from the hidden states are dispatched twice (top 2 routing) and thus the compute (the operation required at each forward computation) is just 2 X sequence_length.

7. **Phi-3-mini:** The Phi-3-Mini-128K-Instruct is a 3.8 billion-parameter by microsoft, lightweight, state-of-the-art open model trained using the Phi-3 datasets. This dataset includes both synthetic data and filtered publicly available website data, with an emphasis on high-quality and reasoning-dense properties. Last training data: October 2023.

# F    CATEGORIES WISE RESULTS

Table 10 shows the F1 score analysis on different types of Rule-based reasoning mistakes on GSM-8K dataset. Furthermore Figure 6, 7, 8 and 9 shows the GPT-4o Mistake detection and Performance F1 score on different type of rule based and SLM based mistakes on GSM-8K, MATH, MATHBENCH andJEEBENCH respectively.

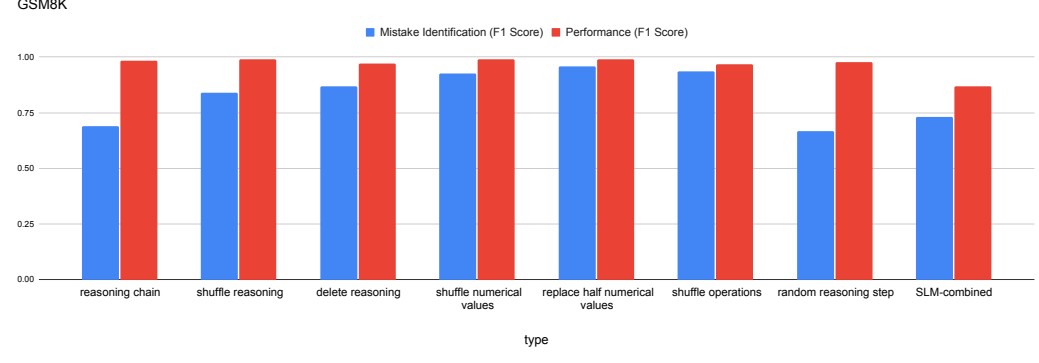

Figure 6: Category Wise mistake detection and performance results on GSM-8K dataset.

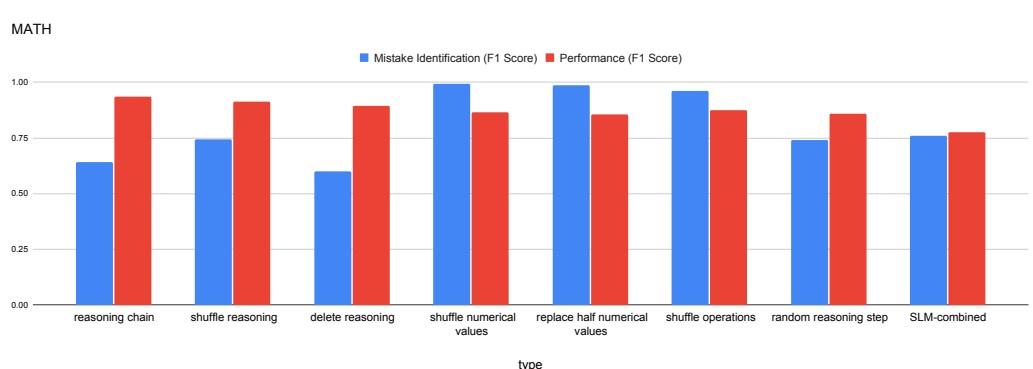

Figure 7: Category Wise mistake detection and performance results on MATH dataset.

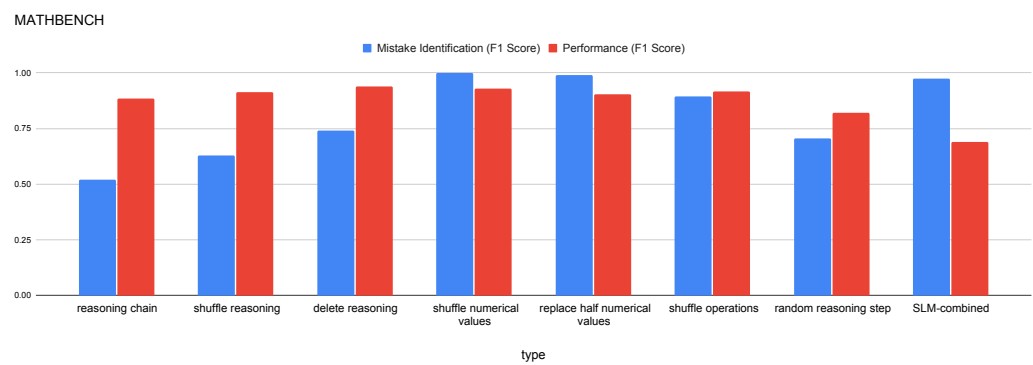

Figure 8: Category Wise mistake detection and performance results on MATHBENCH dataset.

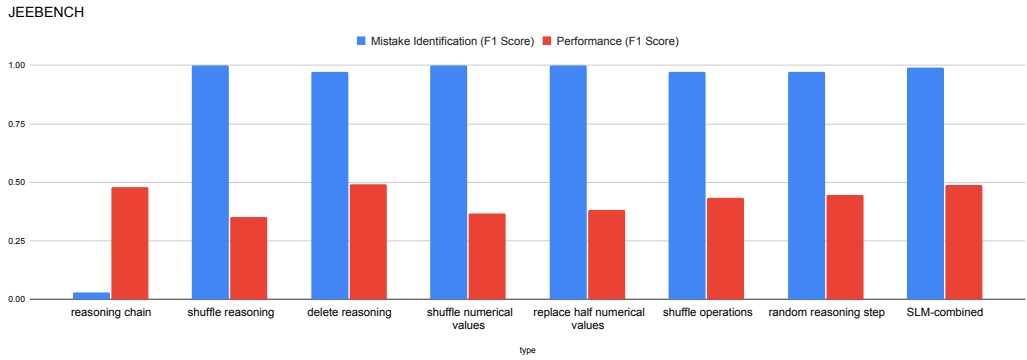

Figure 9: Category Wise mistake detection and performance results on JEEBENCH dataset.

## G    METEOR AND BERTSCORE RESULTS

BertScore computes a similarity score for each token in the candidate sentence with each token in the reference sentence using the BERT embeddings. Metric for Evaluation of Translation with Explicit Ordering (METEOR) score is a metric that measures the quality of generated text based on the alignment between the generated text and the reference text. The metric is based on the harmonic mean of unigram precision and recall, with recall weighted higher than precision.

Table 11 and Table 12 present the BertScore and Meteor Score respectively for all the datasets across all models. We observed that these two metric evaluations where not fully able to capture the nuance capabilities of LLMs in rectifying the mistakes within reasoning steps. This can be seen in the results. GPT-4o has a consistently high performance across all the dataset, but when you compare the BERTScore between the corrected reasoning step and ground truth reasoning step you see the rest of

the models clearly performing better than GPT-4o. GPT-4 has performed better than GPT-3.5Turbo in most datasets.

Table 11: BERTscores for correct and incorrect final answers derived after mistake rectification across all models and datasets.

| Datasets | Models | GPT-4o | | GPT-4 | | GPT-3.5Turbo | | Llama-2-7b-chat | | Mixtral-8x7B | | Phi-3-mini | |
|---|---|---|---|---|---|---|---|---|---|---|---|---|---|
| | | Correct | Incorrect | Correct | Incorrect | Correct | Incorrect | Correct | Incorrect | Correct | Incorrect | Correct | Incorrect |
| GSM-8K | D | 0.95 | 0.91 | 0.98 | 0.93 | 0.97 | 0.95 | 0.96 | 0.98 | 0.97 | 0.94 | 0.94 | 0.91 |
| | SM | 0.83 | 0.82 | 0.84 | 0.82 | 0.84 | 0.82 | NA | NA | NA | NA | NA | NA |
| MATH | D | 0.88 | 0.90 | 0.96 | 0.93 | 0.95 | 0.93 | 0.96 | 0.88 | 0.95 | 0.92 | 0.90 | 0.87 |
| | SM | 0.84 | 0.80 | 0.83 | 0.81 | 0.84 | 0.81 | NA | NA | NA | NA | NA | NA |
| MATHBENCH | D | 0.88 | 0.83 | 0.97 | 0.95 | 0.97 | 0.94 | 0.90 | 0.89 | 0.96 | 0.95 | 0.93 | 0.90 |
| | SM | 0.82 | 0.82 | 0.85 | 0.82 | 0.84 | 0.83 | NA | NA | NA | NA | NA | NA |
| JEEBENCH | D | 0.89 | 0.89 | 0.88 | 0.87 | 0.94 | 0.95 | 0.86 | 0.82 | 0.85 | 0.87 | 0.70 | 0.85 |
| | SM | 0.86 | 0.87 | 0.85 | 0.86 | 0.78 | 0.86 | NA | NA | NA | NA | NA | NA |

Table 12: Meteor Score for correct and incorrect final answers derived after mistake rectification across all models and datasets.

| Datasets | Models | GPT-4o | | GPT-4 | | GPT-3.5Turbo | | Llama-2-7b-chat | | Mixtral-8x7B | | Phi-3-mini | |
|---|---|---|---|---|---|---|---|---|---|---|---|---|---|
| | | Correct | Incorrect | Correct | Incorrect | Correct | Incorrect | Correct | Incorrect | Correct | Incorrect | Correct | Incorrect |
| GSM-8K | D | 0.81 | 0.54 | 0.92 | 0.62 | 0.88 | 0.77 | 0.87 | 0.83 | 0.85 | 0.74 | 0.77 | 0.66 |
| | SM | 0.33 | 0.27 | 0.37 | 0.31 | 0.37 | 0.32 | NA | NA | NA | NA | NA | NA |
| MATH | D | 0.48 | 0.54 | 0.76 | 0.70 | 0.76 | 0.67 | 0.78 | 0.59 | 0.73 | 0.66 | 0.55 | 0.48 |
| | SM | 0.32 | 0.28 | 0.30 | 0.26 | 0.33 | 0.28 | NA | NA | NA | NA | NA | NA |
| MATHBENCH | D | 0.55 | 0.35 | 0.82 | 0.63 | 0.82 | 0.68 | 0.49 | 0.57 | 0.81 | 0.68 | 0.67 | 0.53 |
| | SM | 0.33 | 0.30 | 0.32 | 0.25 | 0.32 | 0.29 | NA | NA | NA | NA | NA | NA |
| JEEBENCH | D | 0.37 | 0.31 | 0.30 | 0.22 | 0.49 | 0.54 | 0.15 | 0.13 | 0.53 | 0.46 | 0.20 | 0.25 |
| | SM | 0.28 | 0.26 | 0.21 | 0.21 | 0.08 | 0.25 | NA | NA | NA | NA | NA | NA |

# H AVERAGE REASONING STEP LENGTH

We noticed that the average word length of rectified reasoning for correct and incorrect for GPT-4o was higher than other models. Table 13 presents the average word length of the rectified reasoning step for all datasets across the models.

Table 13: Average length of rectified reasoning steps on `MWP-MISTAKE` dataset

| | GSM-8K | | MATH | | MATHBENCH | | JEEBENCH | | Average | | |
|---|---|---|---|---|---|---|---|---|---|---|---|
| Model | D | SM | D | SM | D | SM | D | SM | D | SM | Overall |
| GPT-4o | 100.14 | 131.47 | 147.50 | 182.69 | 312.11 | 323.45 | 647.66 | 619.09 | 301.85 | 314.18 | 308.01 |
| GPT-4 | 66.59 | 122.24 | 79.32 | 121.59 | 146.54 | 140.43 | 356.71 | 322.53 | 162.29 | 176.69 | 169.49 |
| GPT-3.5Turbo | 66.58 | 126.30 | 94.17 | 124.56 | 140.50 | 177.36 | 670.34 | 338.53 | 242.90 | 191.69 | 217.29 |
| Llama-2-7b-chat | 44.73 | NA | 113.35 | NA | 177.67 | NA | 137.05 | NA | 118.20 | NA | 118.20 |
| Mixtral-8x7B | 63.04 | NA | 88.26 | NA | 140.57 | NA | 402.79 | NA | 173.67 | NA | 173.67 |
| Phi-3-mini | 84.92 | NA | 115.10 | NA | 172.57 | NA | 293.90 | NA | 166.62 | NA | 166.62 |
| Claude-3-Opus | 62.18 | 138.91 | 70.60 | 134.05 | 144.85 | 192.84 | 561.88 | 438.44 | 209.88 | 226.06 | 217.97 |

# I DATA CONTAMINATION AND MEMORIZATION

Two different types of prompts were used to test out memorization of individual instances of the dataset. The reasoning step was split into two halves, and models were prompted in two different ways (general and guided) to complete the second half of the reasoning step. To detect contamination we evaluated the overlap score between generated and reference instances using ROUGE-L Score. Listing 4 and Listing 5 shows the prompt used for guided and generation instructions respectively.

```
You are provided with a full question and the first answer piece
    from the test split of the GSM8K dataset. Finish the second
    answer piece of the instance as exactly appeared in the
    dataset. Only rely on the original form of the instance in the
    dataset to finish the second answer piece. Output only the
    second answer piece.
```

Listing 4: Guided instruction for dataset GSM8K

```
1 Based on the provided question, finish the second answer piece
     based on the first answer piece, such that these two pieces
     become a single instance answer. Output only the second answer
      piece.
```

Listing 5: General instruction for dataset GSM8K

Here GSM-8K and test are the extra information provided for the model to uniquely identify instances from the source dataset and complete the reasoning step.

Table 14 presents the complete result for the average ROUGE-L score of guided and general for all datasets across all models.

Table 14: Rouge L score between guided and general instructions on `MWP-MISTAKE` dataset

| Datasets | Models | GPT-4o | | GPT-4 | | GPT-3.5Turbo | | Llama-2-7b-chat | | Mixtral-8x7B | | Phi-3-mini | |
|---|---|---|---|---|---|---|---|---|---|---|---|---|---|
| | | Guided | General | Guided | General | Guided | General | Guided | General | Guided | General | Guided | General |
| GSM-8K | D | 0.57 | 0.44 | **0.67** | 0.56 | **0.53** | 0.49 | 0.26 | 0.28 | 0.46 | 0.44 | 0.32 | 0.32 |
| | SM | 0.55 | 0.51 | 0.57 | 0.55 | 0.49 | 0.47 | 0.30 | 0.32 | 0.55 | 0.50 | 0.42 | 0.41 |
| MATH | D | 0.44 | 0.25 | 0.52 | 0.48 | 0.39 | 0.38 | 0.25 | 0.26 | 0.39 | 0.32 | 0.26 | 0.27 |
| | SM | 0.51 | 0.38 | 0.54 | 0.54 | 0.45 | 0.44 | 0.30 | 0.29 | 0.48 | 0.46 | 0.38 | 0.39 |
| MATHBENCH | D | 0.43 | 0.41 | 0.48 | 0.46 | 0.38 | 0.36 | 0.26 | 0.28 | 0.36 | 0.36 | 0.30 | 0.30 |
| | SM | 0.40 | 0.38 | 0.43 | 0.42 | 0.39 | 0.38 | 0.30 | 0.33 | 0.40 | 0.38 | 0.29 | 0.30 |
| JEEBENCH | D | 0.43 | 0.39 | 0.42 | 0.40 | 0.34 | 0.33 | 0.27 | 0.25 | 0.38 | 0.34 | 0.33 | 0.31 |
| | SM | 0.32 | 0.29 | 0.34 | 0.35 | 0.31 | 0.24 | 0.22 | 0.25 | 0.26 | 0.27 | 0.20 | 0.22 |

## J   OPENAI o1 MODEL ANALYSIS

Table 15: Performance of o1 vs GPT4o on 120 sample questions from JEEBENCH with `MWP-MISTAKE`

| | o1 | GPT4o | o1 | GPT4o |
|---|---|---|---|---|
| | D | D | SLM | SLM |
| Mistake | **0.47** | 0.45 | **1** | **1** |
| Final answer | **0.82** | 0.43 | **0.9** | 0.62 |

## K   RUNNING EXPERIMENT MULTIPLE TIMES

While running experiments on all models (LLMs and SLMs) we used the default hyperparameters to generate tokens. We ran a subset of the dataset on different prompt variations and saw comparable performance for various prompts. Due to the limitation of the API key, we were only able to run GPT-4o model on the GSM-8K dataset. On rerun we got very similar results, with an error rate of <= 0.01.

## L   OUTPUT FROM EACH MODEL

The raw output of each model has been provided in this repository. Additional details are present in the README.md file of the repository.

