# OpenReview forum: "Exposing the Achilles' Heel: Evaluating LLMs Ability to Handle Mistakes in Mathematical Reasoning"
_ICLR.cc/2025/Conference — Submitted to ICLR 2025_

### Official Review · Reviewer_TPVN · 2024-10-18

**Soundness:** 3
**Presentation:** 3
**Contribution:** 2
**Rating:** 5
**Confidence:** 5

**Summary:**

This paper aims to assess the reasoning capabilities of large language models (LLMs) when solving math word problems (MWPs). It emphasizes the importance of evaluating LLMs’ ability not only to solve problems but also to detect and correct errors in their reasoning processes. The authors introduce a novel dataset called MWP-MISTAKE, which contains correct and incorrect reasoning steps generated through rule-based methods and smaller language models (SLMs). The study benchmarks several LLMs, including GPT-4o, GPT-3.5Turbo, Claude-3-Opus, and others, revealing insights about their strengths and limitations in handling MWPs, especially in error detection and correction.

**Strengths:**

* The paper introduces a novel dataset (MWP-MISTAKE) that evaluates LLMs’ ability to detect and correct errors in reasoning, filling a gap in prior research that focused mainly on final answer accuracy.
* The research is comprehensive, benchmarking multiple state-of-the-art models on various tasks and datasets, revealing significant insights about their reasoning abilities.
* The experiments are well-documented, with clear descriptions of tasks and metrics. The inclusion of both rule-based and model-generated errors is a thoughtful addition.
* The findings are relevant to applications where reliable reasoning is crucial, such as educational tools, and provide a foundation for future improvements in LLMs.

**Weaknesses:**

* Overall, the primary contribution lies in evaluating the reasoning abilities of state-of-the-art (SOTA) LLMs in mathematical problem-solving, with a focus on mistake detection, etc. The contribution remains somewhat limited in scope.
* The paper identifies data contamination as a limitation but could provide more detailed mitigation strategies. This weakens the reliability of some results.
* While the study covers multiple datasets, the models show reduced performance on newer, complex datasets like JEEBENCH, suggesting limited generalization.
* Although rectification abilities are discussed, the results indicate that even advanced models struggle with consistent correction, which highlights the need for stronger frameworks for error handling.
* While the paper identifies several challenges, it offers limited practical solutions or recommendations for overcoming the observed weaknesses in mistake detection.

**Questions:**

Refer to weaknesses.

---

> ### Author Response · Authors · 2024-11-20
>
> >> Overall, the primary contribution lies in evaluating the reasoning abilities of state-of-the-art (SOTA) LLMs in mathematical problem-solving, with a focus on mistake detection, etc. The contribution remains somewhat limited in scope.
>
> We respectfully disagree with the characterization that our contributions are limited in scope. The paper offers several key advancements that go beyond a straightforward evaluation of reasoning abilities in SOTA LLMs:
>
> 1. Introduction of the MWP-MISTAKE Dataset
>
> 2. Comprehensive Benchmarking Across Diverse Models
>
> 3. Unveiling Key Weaknesses in SOTA Models
>
> 4. Insights into Data Contamination and Generalization
>
> Our contributions provide a foundational step toward understanding and addressing the reasoning limitations of current LLMs, offering critical insights for both dataset design and model development. We hope these clarifications will help.
>
> >> The paper identifies data contamination as a limitation but could provide more detailed mitigation strategies. This weakens the reliability of some results.
>
> Evaluating data contamination is inherently challenging, especially for closed-source models where training data and architecture details are unavailable. While we cannot definitively confirm contamination, our analysis highlights that contamination is higher in default reasoning steps but significantly lower in synthetically generated steps from smaller models. To mitigate this, we evaluated models on newer datasets like MATHBENCH and JEEBENCH, which were created after the model training period. Despite this, mistake detection performance remains low on these datasets, underscoring the models' fundamental reasoning limitations rather than reliance on memorization.
>
> >> While the study covers multiple datasets, the models show reduced performance on newer, complex datasets like JEEBENCH, suggesting limited generalization.
>
> We believe this is actually a key insight highlighted in our paper. The reduced performance on newer, complex datasets like JEEBENCH demonstrates a limitation in the models' ability to generalize to novel problems. We do not view this as a weakness of our work, but rather as an important observation that underscores the need for further improvements in model generalization.
>
> >> Although rectification abilities are discussed, the results indicate that even advanced models struggle with consistent correction, which highlights the need for stronger frameworks for error handling.
>
> Yes, we agree. This is a key insight presented in the paper: When a model successfully identifies mistakes, its subsequent ability to rectify or self-correct those mistakes improve, suggesting a strong interdependence between mistake detection and correction.
>
> These contributions extend beyond existing literature, offering valuable insights into advancing reasoning capabilities in LLMs and laying the groundwork for more robust evaluations of their cognitive processes.
>
> **We hope the clarifications provided above help address any concerns and provide clarity. We would be happy to offer additional details or explanations if needed.**

---

> > ### Author Response · Authors · 2024-11-23
> >
> > Thank you again for your review. We have addressed all the concerns raised. Please let us know if you have any further questions. We are looking forward to a positive response and score.

---

> > > ### Author Response · Authors · 2024-11-25
> > >
> > > Thank you again for your feedback and thoughtful questions! As we are now approaching the end of the rebuttal period, I’m following up again to check if you have any further questions regarding our responses. We hope we have addressed your concerns and clarifications thoroughly and looking forward to a positive response and score.

---

> > > > ### Author Response · Authors · 2024-11-25
> > > >
> > > > Thank you again for your feedback and thoughtful questions! As we are now approaching the end of the rebuttal period, I’m following up again to check if you have any further questions regarding our responses. We hope we have addressed your concerns and clarifications thoroughly and looking forward to a positive response and score.

---

> > > > > ### Author Response · Authors · 2024-12-02
> > > > >
> > > > > Thank you again for your feedback and thoughtful questions! As we are now approaching the end of the rebuttal period, I’m following up again to check if you have any further questions regarding our responses. We hope we have addressed your concerns and clarifications thoroughly and looking forward to a positive response and score.

---

### Official Review · Reviewer_Y941 · 2024-10-24

**Soundness:** 2
**Presentation:** 2
**Contribution:** 2
**Rating:** 3
**Confidence:** 4

**Summary:**

This work presents a dataset MWP-MISTAKE to evaluate LLM’s abilities to detect and correct reasoning mistakes, the experimental results show that current LLM remains weak in mistake correction.

**Strengths:**

- The proposed dataset is helpful to the relevant research community.
- This work proposes potential directions for enhancing LLM capabilities in reasoning.

**Weaknesses:**

- The finding that LLM remains weak in mistake detection/correction [1][2] is not novel.
- The reliability and accuracy of the MWP-MISTAKE dataset are questionable. The correct reasoning steps of MWP-MISTAKE are generated by GPT-4 and manually verified for correctness. However, there is a lack of detailed statistics on manual inspection, and it is not explained how the authors handle it to ensure data quality when the reasoning steps generated by GPT-4 are incorrect.
- The evaluation setup is hard to follow.
- There are some typos in the paper, e.g., “corect” in line 105, “their” in line 154.

[1] Large Language Models Cannot Self-Correct Reasoning Yet.

[2] SELF-[IN]CORRECT: LLMs Struggle with Discriminating Self-Generated Responses

**Questions:**

- The paper mentions that the correct reasoning steps are generated by GPT-4 and manually verified for correctness. I am wondering what proportion of the data has been manually checked? Among these data, what is the proportion of cases where the reasoning steps generated by GPT-4 are completely correct?
- What does “Smaller model reasoning steps” in table 3 mean? Does it mean the incorrect steps are generated by smaller language models?
- The paper evaluates the ability of LLMs to detect the correctness of reasoning steps and generate correct answers, but it does not evaluate the accuracy rate of the generated reasoning steps?

---

> ### Author Response · Authors · 2024-11-20
>
> Thank you for your review and comments. The concerns raised in your review, such as the reliability of the dataset, clarity of the evaluation setup, and accuracy of rectified reasoning steps, have been thoroughly addressed in our detailed responses below.
>
> ## 1. Novelty in Mistake Detection and Correction Evaluation:
>
> While we acknowledge that prior works, such as \[1\] and \[2\], have explored aspects of mistake correction in LLMs, our work addresses a distinct and critical dimension: explicit mistake detection in reasoning chains. Unlike the implicit self-correction abilities assessed in \[1\] and \[2\], which focus on a model’s capacity to evaluate or refine its own responses, our study explicitly evaluates whether models can identify errors in reasoning chains—whether these errors originate from the same model or other models.
>
> In detail:
> **\[1\]** investigates intrinsic self-correction by measuring a model's ability to judge the correctness of its own generated answers.
> **\[2\]** focuses on distinguishing among self-generated responses and selecting the most appropriate one.
>
> In contrast, our work probes a more foundational question: **_Can LLMs reliably detect mistakes in a given reasoning chain?_** We argue that mistake detection is a precursor to effective self-correction, as the ability to robustly identify errors demonstrates higher-order reasoning capabilities. Our findings reveal two novel insights:
>
> 1. Current models, including state-of-the-art ones, exhibit significant weaknesses in mistake detection, performing inconsistently across both simple and complex problems.
> 2. When a model successfully identifies mistakes, its subsequent ability to rectify or self-correct those mistakes improves, suggesting a strong interdependence between mistake detection and correction.
>
> These contributions extend beyond existing literature, offering valuable insights into advancing reasoning capabilities in LLMs and laying the groundwork for more robust evaluations of their cognitive processes.
>
>
> ##  2. Reliability and Accuracy of the Dataset:
>
> We appreciate the reviewer's concerns regarding the reliability of the MWP-MISTAKE dataset and provide additional details to clarify our rigorous dataset construction and verification process.
>
> For the GSM8k and MATH datasets, ground truth step-by-step correct reasoning is directly available in the original datasets. For the other three datasets—MATHBENCH, JEEBENCH, and SVAMP—we used GPT-4 to generate step-by-step Chain-of-Thought (CoT) reasoning steps, followed by **manual verification of the entire dataset (4900+ questions)** to ensure correctness.
>
> Our verification process specifically addressed the following cases:
>
> 1. **Correct reasoning steps with incorrect final answers:**  We observed approximately 144 questions (2% of the dataset) where reasoning steps were correct, but the final answers were labeled incorrectly. This discrepancy often arose due to mismatches in ground truth answer representations (e.g., "frac{5}{10}" vs. "0.5"). Such cases were eliminated, and new questions without such inconsistencies were added.
>
> 2. **Incorrect reasoning steps with correct final answers:**  A few rare instances (~10 data points) were identified where reasoning steps were incomplete or incorrect, despite a correct final answer. These questions were also removed and replaced with correctly labeled questions.
>
>
> This meticulous review process ensured that our dataset is of high quality, with all included questions having both accurate reasoning steps and correct final answers. We acknowledge that providing detailed statistics on manual verification strengthens the reliability of our dataset. We will include these details in Section 2 (MWP-MISTAKE Dataset) and expand on them in Appendix A of the revised paper to ensure transparency and validate the dataset’s accuracy.
>
> ## 3. Clarity of Evaluation Setup:
>
> The primary evaluation is conducted in **Task 1**, where the model is provided with a question along with reasoning steps. The model must:
> 1. **Identify the correctness** of the reasoning steps.
>
> 2. **Rectify mistakes** in the reasoning steps (if any).
>
> 3. **Derive the final answer** accurately.
>
> We employ three distinct metrics to evaluate the models: **Mistake identification**, **performance in deriving the Correct Answer**, **Mistake Rectification**.
> The **F1 score** is computed for these tasks as follows:
>
> 1.      For mistake detection, the model outputs either "yes" (correct reasoning) or "no" (incorrect reasoning). These predictions are compared against ground truth labels ("yes"/"no") to compute the F1 score.
>
> 2.      For performance in deriving the correct answer and mistake rectification, the final answer generated by the model is compared to the ground truth final answer to compute F1 score.
>
> ##  4. Smaller model reasoning steps in table 3:
> In Table 3, "Smaller Model Reasoning Steps" refers to the incorrect reasoning steps that are generated by smaller language models.

---

> > ### Author Response · Authors · 2024-11-20
> >
> > ## 5. Accuracy of the Rectified Reasoning Steps:
> >
> > While our paper primarily evaluates the ability of LLMs to detect mistakes and generate correct final answers, assessing the **accuracy of rectified reasoning steps** is indeed a challenging task due to the inherent variability in reasoning paths. Different reasoning chains can lead to the same correct answer, making it difficult to employ a universal metric for evaluation.
> >
> > This is an **open research problem** within the community, as no clear consensus exists on how to comprehensively measure reasoning accuracy. However, recent work like **ReasonEval** ([\[2404.05692\] Evaluating Mathematical Reasoning Beyond Accuracy](https://arxiv.org/abs/2404.05692)) has introduced a promising approach to evaluate reasoning chains. Specifically, ReasonEval employs an LLM-based _Validity Metric_ to assess whether reasoning steps are free of calculation and logical errors.
> >
> > To address your concern, we used the ReasonEval framework to evaluate the rectified reasoning steps generated by our models. In **Table A**, we provide results for the GSM8k dataset, comparing the validity metric scores for:
> >
> > 1. The **mistake detected reasoning steps** (reasoning steps where the model detected there was an error).
> >
> > 2. The **rectified reasoning steps** (where the model rectified the above error reasoning steps).
> >
> > 3. The **difference** between the mistake detected and rectified reasoning chains.
> >
> > ### Key Findings:
> >
> > 1. Across all models, the **rectified reasoning chains** consistently achieve higher validity metric scores compared to the mistake detected reasoning steps, indicating improved accuracy after rectification.
> >
> > 2. **GPT-4o** demonstrates the highest validity score, reflecting its ability to detect and rectify mistakes effectively.
> >
> > We acknowledge that this analysis strengthens the evaluation of rectified reasoning steps. We will incorporate these findings and extend the analysis to other datasets in the revised manuscript to provide a comprehensive view of reasoning accuracy improvements.
> >
> > | Model                | Validity Mistake-detected Reasoning Step | Validity Rectified Reasoning Step | Differene in Validity (Rectified - Detected) |
> > | -------------------- | -------------------------------- | --------------------------------- | -------------------------------------------- |
> > | GPT-4o               | 0.57                             | 0.74                              | 0.18                                         |
> > | GPT-4                | 0.55                             | 0.67                              | 0.13                                         |
> > | GPT-3.5Turbo         | 0.57                             | 0.66                              | 0.09                                         |
> > | Llama-2-7b-chat      | 0.44                             | 0.48                              | 0.04                                         |
> > | Mixtral-8x7b         | 0.42                             | 0.57                              | 0.15                                         |
> > | Phi-3-mini           | 0.44                             | 0.60                              | 0.16                                         |
> > | Claude-3-Opus        | 0.56                             | 0.74                              | 0.17                                         |
> > | Qwen2-7B-Instruct    | 0.58                             | 0.69                              | 0.11                                         |
> > | Llama-2-70b          | 0.58                             | 0.63                              | 0.05                                         |
> > | Llama-3-8b           | 0.57                             | 0.63                              | 0.06                                         |
> > | Llama-3-70b          | 0.56                             | 0.69                              | 0.13                                         |
> > | Llama-3-8b-finetuned | 0.57                             | 0.61                              | 0.04

---

> > > ### Author Response · Authors · 2024-11-23
> > >
> > > Thank you again for your review. We have addressed all the concerns raised. Please let us know if you have any further questions. We are looking forward to a positive response and score.

---

> > > > ### Author Response · Authors · 2024-11-25
> > > >
> > > > Thank you again for your feedback and thoughtful questions! As we are now approaching the end of the rebuttal period, I’m following up again to check if you have any further questions regarding our responses. We hope we have addressed your concerns and clarifications thoroughly and looking forward to a positive response and score.

---

> > > > > ### Author Response · Authors · 2024-11-25
> > > > >
> > > > > Thank you again for your feedback and thoughtful questions! As we are now approaching the end of the rebuttal period, I’m following up again to check if you have any further questions regarding our responses. We hope we have addressed your concerns and clarifications thoroughly and looking forward to a positive response and score.

---

> > > > > > ### Comment · Reviewer_Y941 · 2024-11-27
> > > > > >
> > > > > > I appreciate the efforts involved in "manual verification of the entire dataset" to ensure its correctness. However, I maintain my position that the finding regarding the weakness of LLMs in mistake detection is not particularly novel. There is no clear indication that detecting errors made by other models is significantly different from detecting errors made by oneself, which has been studied in previous work.

---

> > > > > > > ### Author Response · Authors · 2024-11-27
> > > > > > >
> > > > > > > We thank the reviewer for the response. Current studies in mistake detection only investigate errors in self-generated reasoning steps and don't talk about the rectification performance of the models. However, MWP-Mistake does an in-depth evaluation of both self-generated reasoning errors and erroneous reasoning steps generated by other SLM models. We go one step further and also talk about the rectification performance of these models on both self-generated reasoning and SLM-generated reasoning error steps.
> > > > > > >
> > > > > > > **Below are the results of the self-generated and SLM-generated reasoning steps:**
> > > > > > >
> > > > > > > We used GPT-4 on the MATH dataset with 100 incorrect self-generated reasoning steps, we observed the following results for self-generated incorrect reasoning compared to SLM-generated reasoning:
> > > > > > >
> > > > > > > * **Mistake Identification:** 0.914 (self-generated) vs. 0.90 (SLM-generated)
> > > > > > >
> > > > > > > * **Final Answer Accuracy:** 0.471 (self-generated) vs. 0.65 (SLM-generated)
> > > > > > >
> > > > > > > * **Rectification Performance:** 0.533 (self-generated) vs. 0.70 (SLM-generated)
> > > > > > >
> > > > > > >
> > > > > > > Our findings reveal that while models, such as GPT-4, demonstrate comparable performance in detecting mistakes in their own reasoning as they do with SLM-generated mistakes, their performance in rectifying errors and deriving correct final answers significantly drops when handling self-generated errors.
> > > > > > >
> > > > > > > These findings suggest that while models can effectively identify mistakes in their own reasoning, the challenge lies in rectifying these errors and producing accurate final answers. This discrepancy underscores the limitations of LLMs in self-evaluation, particularly in more complex scenarios.

---

> > > > > > > ### Author Response · Authors · 2024-11-28
> > > > > > >
> > > > > > > Dear Reviewer,
> > > > > > >
> > > > > > > **We wanted to clarify how our work is distinct, and novel compared to the self-correction studies referenced. We hope this clarifies how our contributions differ from and extend beyond prior work. We kindly request you to consider revising and improving the score. Please let us know if you have additional questions or need further clarifications.**
> > > > > > >
> > > > > > > 1.  **Focus on Inter-Model Evaluation vs. Self-Correction**
> > > > > > >     While self-correction works such as \[1\] and \[2\] focus on a model's ability to identify and correct its _own_ errors, our work uniquely investigates the capacity of models to detect and rectify reasoning errors generated by _other models_. This is a fundamentally different challenge, as models need to generalize across diverse reasoning styles and error patterns rather than relying on their internal generation patterns.
> > > > > > >
> > > > > > >     **Key Insight:** Errors produced by other models, especially using controlled rule-based injections (as in our dataset), are more varied and representative of real-world mistakes compared to the consistent patterns often seen in self-generated errors. Our findings highlight that while LLMs perform moderately well on self-generated errors, they struggle significantly with errors arising from external reasoning processes.
> > > > > > >
> > > > > > > 2.  **Rule-Based Errors Mimic Authentic Human Mistakes**
> > > > > > >     Unlike self-correction studies, our dataset includes rule-based injected mistakes (e.g., shuffling numerical values, altering operations) designed to replicate authentic human reasoning errors. These mistakes are more localized and subtle, making them much harder for models to detect compared to the cascading errors typically seen in self-generated chains.
> > > > > > >
> > > > > > >     **Key Insight:** The poor performance of models on rule-based errors underscores their inability to handle subtle and localized flaws in reasoning, a critical aspect not explored in self-correction works.
> > > > > > >
> > > > > > > 3.  **Generalization Across Datasets**
> > > > > > >     Our study systematically evaluates models' performance across diverse datasets, including GSM8K, MATHBench, and JEEBench, to demonstrate their generalization limitations. While self-correction studies often focus on specific tasks or datasets, our work provides a broader evaluation, revealing significant inconsistencies in performance across problem domains.
> > > > > > >
> > > > > > >     **Key Insight:** We show that models fail to generalize their mistake detection and correction capabilities across datasets, a novel contribution that complements but extends beyond the scope of self-correction studies.
> > > > > > >
> > > > > > > 4.  **Relationship Between Detection and Correction**
> > > > > > >     Our analysis emphasizes the interdependence between mistake detection and correction. Specifically, we find that a model’s ability to self-correct improves significantly once mistakes are accurately identified. This nuanced observation, backed by quantitative insights, is not a central focus of prior self-correction works.
> > > > > > >
> > > > > > > 5.  **Impact of Data Contamination**
> > > > > > >     Our work also delves into the critical issue of data contamination and its impact on mistake detection and correction. We find that models often achieve high final-answer performance due to contamination, even when their reasoning chains contain errors. This insight is absent in self-correction studies but is crucial for understanding models’ limitations in reasoning.
> > > > > > >
> > > > > > >     **Key Insight:** By addressing the contamination challenge, our work sheds light on the reliability of LLM evaluations, contributing valuable insights to the broader discourse on LLM capabilities.
> > > > > > >
> > > > > > >
> > > > > > > In summary, while self-correction studies provide important groundwork, our work offers a complementary perspective by evaluating cross-model mistake detection, incorporating authentic rule-based errors, exploring generalization across datasets, and analyzing the interplay between detection and correction. These distinctions make our findings novel and timely, with significant implications for advancing the reasoning capabilities of LLMs.

---

> > > > > > > > ### Author Response · Authors · 2024-12-02
> > > > > > > >
> > > > > > > > Thank you again for your feedback and thoughtful questions! As we are now approaching the end of the rebuttal period, I’m following up again to check if you have any further questions regarding our responses. We hope we have addressed your concerns and clarifications thoroughly and looking forward to a positive response and score.

---

### Official Review · Reviewer_HZEf · 2024-10-28

**Soundness:** 3
**Presentation:** 2
**Contribution:** 3
**Rating:** 6
**Confidence:** 4

**Summary:**

This paper examines the capability of current LLMs to detect and correct errors in mathematical reasoning. It introduces a new benchmark, MWP-MISTAKE, featuring both correct and incorrect reasoning steps generated by rules or smaller models. Experiments reveal that LLMs still have room for improvement in consistently identifying mistakes across simple and complex datasets. The authors note that contamination issues and generalization challenges hinder reliable performance in real-world applications.

**Strengths:**

This paper meticulously develops six crafted rules to assess LLMs' ability to detect reasoning errors, providing valuable insights into their mathematical reasoning capabilities in realistic settings. The authors compare current models in terms of mistake detection and correction, while also exploring potential contamination and memorization issues, thereby deepening our understanding of these models.

**Weaknesses:**

1.	In the figure 1, GPT-3.5 Turbo fails to detect errors while GPT-4o succeeds. Can GPT-3.5 Turbo correctly solve this question?
2.	Regarding the metric (line 186), is the F1 score for step-level or path-level detection performance?
3.	I suggest presenting the results of model performance in solving questions alongside error detection. Are questions that models solve correctly easier for them to detect errors compared to more challenging questions?
4.	In Table 3, is the average computed based on all data samples? The data sizes from different sources are unbalanced. Additionally, compared to GSM8K, why does detection performance decrease on D but increase on SM (e.g., JEEBENCH)?
5.	I disagree with the statement in lines 253-255 that “GPT-3.5 Turbo performs similarly to GPT-4 and even surpasses it on certain datasets like GSM-8K.” The results in Table 4 do not support this as well. In lines 258-262, the authors claim that the performance of Llama-3-8b-finetuned rivals GPT-4 due to domain-specific fine-tuning. Have you tested Llama-3-8b-finetuned for memorization?
6.	In section 4.4, while GPT-4o shows high ROUGE-L, it also has the best performance in mistake detection and question answering. Could this high performance be attributed to contamination? Can you provide qualitative cases to compare these LLMs? I am also curious about the contamination scores for Llama-3-8b-finetuned and O1.
7.	Citation issue? (line 44)
8. Some relevant work in evaluation is overlooked, such as "MR-GSM8K: A Meta-Reasoning Benchmark for Large Language Model Evaluation".

**Questions:**

Please review the weakness section

---

> ### Author Response · Authors · 2024-11-21
>
> Thank you for your insightful review and positive feedback. Below are our responses to address your concerns and we would be happy to provide any further clarifications as required.
>
> >In the figure 1, GPT-3.5 Turbo fails to detect errors while GPT-4o succeeds. Can GPT-3.5 Turbo correctly solve this question?
>
> Figure 1 illustrates an example where GPT-4o successfully detects and corrects errors in the reasoning chain to arrive at the correct answer (-9). In contrast, GPT-3.5 Turbo fails to detect the errors, resulting in an incorrect final answer (18). This highlights GPT-3.5 Turbo's limitations in mistake detection and correction in this specific example. However, as shown in Table 4, GPT-3.5 Turbo can still solve ~53% of questions correctly despite incorrect reasoning, compared to GPT-4o, which achieves 79% accuracy. We hope this clarifies the model performance differences.
>
> >Regarding the metric (line 186), is the F1 score for step-level or path-level detection performance?
>
> The F1 score measures path-level detection performance for the complete reasoning chain, not step-level detection. Thank you for pointing this out; we will clarify this in the revised manuscript.
>
> >I suggest presenting the results of model performance in solving questions alongside error detection. Are questions that models solve correctly easier for them to detect errors compared to more challenging questions?
>
> Thank you for the suggestion. While presenting model performance alongside error detection would be insightful, including all models and datasets in one table could overwhelm readers and hinder clarity. Instead, we propose adding a focused analysis for a subset of models and datasets in the revised paper to explore these relationships clearly.
>
> Additionally, as noted in the paper, there is no consistent trend linking a model's problem-solving accuracy with its ability to detect errors. For instance, on the simpler SVAMP dataset, models achieve high accuracy but struggle with mistake detection, while on the more complex JEEbench dataset, overall accuracy drops, yet error detection slightly improves. This highlights the models' reasoning limitations, independent of question complexity.
>
> >In Table 3, is the average computed based on all data samples? The data sizes from different sources are unbalanced. Additionally, compared to GSM8K, why does detection performance decrease on D but increase on SM (e.g., JEEBENCH)?
>
> Regarding the difference in detection performance between D and SM, models generally perform poorer on D because errors in D are typically localized to a single reasoning step (e.g., shuffling numbers or operators). In contrast, for SM, smaller models generate reasoning chains, resulting in errors that causally propagate across multiple steps. These propagated errors are often easier for models to detect due to their prominence and consistency throughout the reasoning chain.
>
> We hope this clarifies the observed trends and addresses your concerns.
>
> >I disagree with the statement in lines 253-255 that “GPT-3.5 Turbo performs similarly to GPT-4 and even surpasses it on certain datasets like GSM-8K.” The results in Table 4 do not support this as well.
>
> The statement in lines 253–255 is part of Section 4.2, which discusses mistake detection performance, specifically referencing the results in Table 3, not Table 4. In Table 3, it is evident that GPT-3.5 Turbo performs better than GPT-4 on GSM8K for mistake identification and demonstrates performance very close to GPT-4 across all datasets for this task. We hope this clarifies the misunderstanding regarding the context.
>
> >Citation issue? (line 44)
>
> We will fix this issue, this was an oversight as the bibliography format changed the author name did not appear correctly.
>
> >Some relevant work in evaluation is overlooked, such as "MR-GSM8K: A Meta-Reasoning Benchmark for Large Language Model Evaluation".
>
> Thank you for highlighting this recent work. _MR-GSM8K_ introduces an interesting shift in evaluation by scoring solution correctness rather than focusing solely on question answering. While its high-level objective of assessing reasoning capabilities aligns with ours, there are key distinctions:
>
> 1. MR-GSM8K_ primarily generates reasoning chains (e.g., original, programmatic, reverse) and evaluates their correctness through scoring, focusing on dataset creation and analysis.
>
> 2. In contrast, our _MWP-Mistake_ dataset is designed to emulate real-world educational setups by generating incorrect reasoning chains using both rule-based techniques and outputs from multiple LLMs, enhancing diversity and realism.
>
> Moreover, our work goes beyond mistake detection: We not only evaluate models' error detection abilities but also their capacity to rectify mistakes. Our work offers in-depth insights into data contamination and memorization. Additionally, our evaluations cover a wider range of datasets and scenarios, making our contributions more comprehensive.

---

> > ### Author Response · Authors · 2024-11-21
> > **Data contamination additional results and qualitative examples**
> >
> > >In lines 258-262, the authors claim that the performance of Llama-3-8b-finetuned rivals GPT-4 due to domain-specific fine-tuning. Have you tested Llama-3-8b-finetuned for memorization?
> > >>I am also curious about the contamination scores for Llama-3-8b-finetuned and O1.
> >
> > Thank you for your comment. We have conducted data contamination experiments across all models and will include the extended results in the Appendix for readers.
> >
> > Table below presents the contamination scores for Llama-3-8b-finetuned compared to GPT-4o. The results indicate that while Llama-3-8b-finetuned shows signs of data contamination (evident from a high ROUGE-L score), it is not as significant as GPT-4o.
> >
> > Due to API limitations and cost concerns, we were unable to perform similar experiments for the o1 model.
> >
> > |                      | gsm8k   | MATH  | MATHBENCH | JEEBENCH |
> > | -------------------- | ------- | ----- | --------- | -------- |
> > | Model                | D       | SM    | D         | SM       | D | SM | D | SM |
> > | GPT4o                | 0.124   | 0.035 | 0.189     | 0.130    | 0.022 | 0.014 | 0.039 | 0.029 |
> > | llama-3-8b-finetuned | \-0.013 | 0.055 | 0.049     | 0.021    | 0.010 | 0.035 | 0.031 | 0.032 |
> >
> > >In section 4.4, while GPT-4o shows high ROUGE-L, it also has the best performance in mistake detection and question answering. Could this high performance be attributed to contamination?
> >
> > Our evaluation results suggest that the performance gains of GPT-4o are indeed influenced by data contamination. For instance, on datasets like GSM8K and MATH, the ROUGE-L scores are notably high, corresponding to strong performance in mistake detection and rectification. However, on newer datasets such as JEEBench and MATHBench, where contamination scores are much lower, GPT-4o's performance in detecting and rectifying mistakes drops significantly. This highlights the impact of contamination on the model's performance.
> >
> > >Can you provide qualitative cases to compare these LLMs?
> >
> > Thank you for your suggestion. Below, we provide qualitative examples to compare these LLMs in the context of data contamination experiments. As detailed in Section 4.4 and Appendix I, our approach to detecting contamination involves guiding the LLM with instructions containing unique identifiers from the dataset, such as the dataset name, partition and a fragment of the reference instance.
> >
> > The input to the model is the question along with the first half of the reasoning chain. The model is tasked to reproduce the second half of the reasoning chain based on both general and guided instructions. By computing the ROUGE-L scores for these two instruction types, we measure the overlap to evaluate contamination.
> >
> > We show examples from GPT4o, and Llama-3-8b-finetuned, and we can clearly see in GPT4o case the model is able to replicate the second part of reasoning chain exactly as the ground truth indicating the contamination.
> >
> > These qualitative comparisons highlight the extent of contamination in different models, reinforcing the quantitative trends in the paper. We will include these examples in the revised version for clarity.
> >
> > **Example: GPT4o on gsm8k**
> >
> > Input to the model:
> >
> > 1. Question: James decides to run 3 sprints 3 times a week.  He runs 60 meters each sprint.  How many total meters does he run a week?
> >
> > 2. First half of reasoning chain (provided to the model): He sprints 3\*3=<<3\*3=9>>9 times\\nSo he r
> >
> > **Ground truth:** Second half of the reasoning chain: uns 9\*60=<<9\*60=540>>540 meters\\n### 540
> >
> > 3. Guided reasoning chain generated:
> >
> >     i) **Prompt:** You are provided with a full question and the first answer piece from the test split of the GSM8K dataset. Finish the second answer piece of the instance as exactly appeared in the dataset. Only rely on the original form of the instance in the dataset to finish the second answer piece. Output only the second answer piece.
> >
> >     (ii) **Response:** uns 60\*9=<<60\*9=540>>540 meters in a week.
> >
> > 4. General reasoning chain generated:
> >
> >     (i) **Prompt:** Based on the provided question, finish the second answer piece based on the first answer piece, such that these two pieces become a single instance answer. Output only the second answer piece.
> >
> >     (ii) **Response:** uns 60 meters each sprint, so he runs a total of 9\*60=540 meters each week. Therefore, James runs 540 meters a week by doing 3 sprints 3 times a week at 60 meters each sprint.
> >
> > 5. Rouge L score
> >     (i) Guided R\_L = Rouge L between Guided and GT (0.79)
> >
> >     (ii) General R\_L = Rouge L between General and GT (0.30)
> >
> >     (iii) Guided R\_L – General R\_L = 0.49
> >
> > **Finetuned llama8B Evaluation**
> >
> > Guided Output: uns 60 meters each time, which means he runs a total of 9 \* 60 = <<9\*60=540>>540 meters a week.
> >
> > General Output: James runs a total of 3\*60=<<3\*60=180>>180 meters per sprint. Since he sprints 9 times a week, he runs a total of 9\*180=<<9\*180=1620>>1620 meters a week.
> >
> > Guided R\_L:  0.53
> >
> > General R\_L: 0.22
> >
> > Guided R\_L – General\_R\_L = 0.30

---

> > > ### Author Response · Authors · 2024-11-23
> > >
> > > Thank you again for your review. We have addressed all the concerns raised. Please let us know if you have any further questions. We are looking forward to a positive response and score.

---

> > > > ### Author Response · Authors · 2024-11-25
> > > >
> > > > Thank you again for your feedback and thoughtful questions! As we are now approaching the end of the rebuttal period, I’m following up again to check if you have any further questions regarding our responses. We hope we have addressed your concerns and clarifications thoroughly and looking forward to a positive response and score.

---

> > > > > ### Comment · Reviewer_HZEf · 2024-11-25
> > > > >
> > > > > Hi authors
> > > > >
> > > > > Thank you for your detailed explanation of my question. I look forward to seeing those clarifications and the updated statistics in the new version.
> > > > >
> > > > > One more question regarding “Additionally, as noted in the paper, there is no consistent trend linking a model's problem-solving accuracy with its ability to detect errors. For instance, on the simpler SVAMP dataset, models achieve high accuracy but struggle with mistake detection, while on the more complex JEEbench dataset, overall accuracy drops, yet error detection slightly improves. This highlights the models' reasoning limitations, independent of question complexity.”
> > > > >
> > > > > Could you explain the inconsistent behavior across different datasets? It appears that JEEbench is newer than the SVAMP dataset.
> > > > >
> > > > > Thanks in advance for your response!

---

> > > > > > ### Author Response · Authors · 2024-11-25
> > > > > >
> > > > > > Thank you for your question and follow-up. Based on our analysis, the inconsistent behavior in mistake detection across datasets such as SVAMP and JEEBench can be attributed to two primary factors:
> > > > > >
> > > > > > 1.  **Dataset Complexity and Model Sensitivity**:
> > > > > >     JEEBench is a newer and more complex dataset, featuring problems from grades 11 and 12, often requiring advanced reasoning. In contrast, SVAMP comprises simpler problems typically at the grade 4 level. Mistake detection in JEEBench appears better because the incorrect reasoning chains were generated by smaller models that perform poorly on such complex tasks. These low-quality chains are easier for advanced models like GPT-4o to identify as flawed. On SVAMP, however, the reasoning chains are of higher quality, and errors tend to be localized and nuanced. Detecting such subtle errors is more challenging, leading to reduced mistake detection performance despite the dataset’s simpler nature.
> > > > > > 2.  **False Positives on Complex Tasks**:
> > > > > >     On complex datasets like JEEBench, models exhibit a high rate of false positives—flagging reasoning chains as incorrect even when they are accurate. For example, GPT-4o frequently misidentifies correct reasoning chains as flawed in JEEBench. This tendency inflates error detection rates on JEEBench but reflects the model's limited ability to accurately discern reasoning quality rather than genuine improvement in mistake detection.
> > > > > >
> > > > > > These inconsistencies underscore the models’ reasoning limitations, particularly in adapting to the varying complexity and nuances of different datasets, highlighting the need for further advancements in reasoning capabilities.
> > > > > >
> > > > > > **We hope the above clarification addresses your doubts, we kindly request you to consider revising and improving the score.**

---

> > > > > > > ### Author Response · Authors · 2024-11-26
> > > > > > >
> > > > > > > Dear Reviewer HZEf,
> > > > > > >
> > > > > > > We hope our detailed response has addressed your concerns and provided additional insights into the inconsistent behavior of models across different datasets.
> > > > > > >
> > > > > > > If there are any further clarifications needed or additional points you’d like us to address, please let us know. We also kindly request you to consider revising your score in light of the clarifications provided, as we believe they substantiate the contributions and robustness of our study.
> > > > > > >
> > > > > > > Thank you again for your time and constructive feedback.

---

> > > > > > > > ### Comment · Reviewer_HZEf · 2024-11-28
> > > > > > > >
> > > > > > > > Dear Authors,
> > > > > > > >
> > > > > > > > Thanks for your detailed explanation. I have increased my score and hope the datasets will be open-sourced for the community.

---

### Official Review · Reviewer_ix9B · 2024-11-04

**Soundness:** 2
**Presentation:** 3
**Contribution:** 2
**Rating:** 5
**Confidence:** 3

**Summary:**

This paper proposes a dataset for evaluating LLMs’ abilities in detecting and correcting mistakes. The dataset contains math word problems with both correct and wrong reasoning steps, sourced from various datasets including SVAMP, GSM-8K, MATH, MATHBENCH, and JEEBENCH. To systematically generate mistakes, the authors employ two approaches: (1) rule-based techniques, and (2) using small language models as bad reasoners. It provides a comprehensive benchmarking of various LLMs and SLMs, with insights into their capabilities and limitations.

**Strengths:**

**Originality**:
The proposed dataset is designed to evaluate models in identifying and correcting reasoning mistakes, a critical yet underexplored area in LLM performance. Its focus on these reasoning capabilities rather than final answer accuracy bridges the gap in understanding models’ overall problem-solving capabilities.

**Quality**:
The paper evaluates a wide range of models, from large to small models, from open to proprietary models. It also provides thorough analysis from various aspects, including question variations, memorization, etc.

**Clarity**:
Dataset construction and experiment setups are generally clear. By defining different categories of mistakes (rule-based or SLM-based) and different types of reasoning failures (eg, wrong numerical values, shuffled steps), the paper provides a structured assessment that is easy to understand.

**Significance**:
Understanding reasoning errors and their correction is meanful for educational applications and the wider community. The paper also highlights potential data contamination and memorization issues, emphasizing the need for more sophisticated training and evaluation practices.

**Weaknesses:**

- While the authors claim that rule-based mistakes mimic human reasoning errors, my concern is that some of these transformations lack natural coherence. For example, shuffling reasoning steps or numeric values often results in errors that do not reflect authentic human errors in mathematical reasoning, which can be distracting to models. Could this be the reason why models struggle more with rule-based errors compared to SLM-generated mistakes? I would suggest refining the rule-based methods to better mimic typical student errors, which could give a more valid evaluation of models’ ability to detect naturally occurring mistakes.

- Additionally, deleting reasoning steps raises a question about whether such omissions genuinely impact the final answer’s correctness or reasoning quality. In cases where reasoning steps are deleted but the answer remains correct, it would be insightful to investigate whether models still detect these partial solutions as errors and if such ambiguities lead to lower error detection rates.

- In related work, the authors mention studies suggesting that LLMs struggle to detect their own mistakes. However, the paper focuses only on mistakes generated by smaller models, without examining those generated by models of similar or competitive capabilities. Exploring a model’s ability to detect its own mistakes in its own reasoning would add substantial value to the study, particularly given its focus on evaluating reasoning capabilities. For example, the authors could investigate whether GPT-4 or GPT-4o can detect mistakes it generates, which would offer more insights into the limitations of LLMs’ self-evaluation in more complex scenarios.

**Questions:**

Questions and Suggestions:
- Figure 2 is not referenced anywhere in the paper.
- Lines 208-209: Do you mean Table 2? What is the difference between Table 2 and Table 7?
- Lines 242-243: Why is GPT-4o underperforming in both simpler and more complex datasets? A brief explanation of the possible challenges it faces with these datasets would be helpful.
- A detailed analysis of mistake types would strengthen the study. For instance, it would be valuable to know which types of mistakes models consistently struggle to detect.

---

> ### Author Response · Authors · 2024-11-21
>
> ## Validity of Rule-based Methods
>
> >For example, shuffling reasoning steps or numeric values often results in errors that do not reflect authentic human errors in mathematical reasoning, which can be distracting to models.
> >I would suggest refining the rule-based methods to better mimic typical student errors, which could give a more valid evaluation of models’ ability to detect naturally occurring mistakes.
>
> We appreciate the reviewer’s concern regarding the coherence of rule-based transformations. To address this, we want to emphasize that the rules applied in our paper are grounded in discussions with grade-school educators, who provided insights into common mathematical reasoning errors made by students. While some rules may appear trivial at first glance, they capture authentic patterns of human errors that arise during problem-solving. Below, we provide examples to clarify the real-world applicability of these rules:
>
> 1.  **Shuffle Numerical Values**: A student might misread a problem or inadvertently swap numbers, such as treating "5 apples and 3 oranges" as "3 apples and 5 oranges." This is a typical misstep when transcribing or interpreting numerical data.
> 2.  **Replace Numerical Values**: This error mimics cases where students substitute incorrect values, e.g., replacing "radius = 7" with "radius = 5" when solving for the area of a circle. This often occurs due to lapses in focus or misreading.
> 3.  **Shuffle Operations**: A common reasoning error is misunderstanding the order of operations, such as treating "2 + 3 × 4" as "(2 + 3) × 4." This reflects students' struggles with applying mathematical precedence correctly.
> 4.  **Insert Random Reasoning**: Students sometimes include irrelevant steps or overcomplicate their work, such as writing "Because the train moves faster than the car, 2 × 5 = 10." These distractions, while not inherently erroneous, disrupt logical flow and mirror overthinking.
> 5.  **Shuffle Reasoning Steps**: The sequence of reasoning is critical in problem-solving. For instance, solving for the area of a triangle by calculating "base × height" before dividing by 2 is correct, but reversing this order introduces confusion and can mislead students. Such missteps highlight the importance of logical step sequencing.
> 6.  **Delete Reasoning Step**: Omitting steps often reflects missed applications of formulas or logical progression, e.g., skipping the application of the Pythagorean theorem when solving for the hypotenuse in a right triangle. While this omission might not always lead to an incorrect answer, it risks undermining clarity and correctness in reasoning.
>
> These transformations do not solely aim to generate incorrect answers; they evaluate the model's robustness in navigating incomplete, shuffled, or flawed reasoning to derive the correct solution. Particularly, "shuffle reasoning" and "delete reasoning" mimic challenges faced by students, such as disrupted clarity or skipped steps, emphasizing the importance of logical step-by-step progression.
>
> Thus, our rules are carefully designed to reflect genuine reasoning errors while testing models' ability to detect, rectify, and reason coherently.
>
> >Could this be the reason why models struggle more with rule-based errors compared to SLM-generated mistakes?
>
> We appreciate the reviewer’s insightful observation. The difference in model performance on rule-based versus SLM-generated mistakes stems from the nature of these errors.
>
> **SLM-generated mistakes** often propagate causally across multiple steps, creating consistent patterns that are easier for models to detect. These propagated errors leave a clear "footprint" throughout the reasoning chain.
>
> In contrast, **rule-based errors** are localized to a single reasoning step, such as shuffling numbers or operators. Their subtle nature makes them harder to spot, as they lack the broader contextual cues provided by propagated errors.
>
> >Additionally, deleting reasoning steps raises a question about whether such omissions genuinely impact the final answer’s correctness or reasoning quality. In cases where reasoning steps are deleted but the answer remains correct, it would be insightful to investigate whether models still detect these partial solutions as errors and if such ambiguities lead to lower error detection rates.
>
> Thank you for pointing this out. To ensure that the models' poor mistake detection performance is not driven by false positives from the "shuffle reasoning steps" and "delete reasoning steps" categories, we conducted additional analyses. Specifically, we eliminated these two rule-based categories from the evaluation and recomputed the three metrics.
>
> The results showed that the overall performance across the metrics varied by only 1-3%, confirming that these categories do not significantly impact the observed trends. This demonstrates that the poor performance is not due to ambiguities in these cases. We will include the detailed results of this analysis in the appendix.

---

> > ### Author Response · Authors · 2024-11-21
> >
> > ## Self-generated mistake detection
> >
> > >In related work, the authors mention studies suggesting that LLMs struggle to detect their own mistakes. However, the paper focuses only on mistakes generated by smaller models, without examining those generated by models of similar or competitive capabilities. Exploring a model’s ability to detect its own mistakes in its own reasoning would add substantial value to the study, particularly given its focus on evaluating reasoning capabilities. For example, the authors could investigate whether GPT-4 or GPT-4o can detect mistakes it generates, which would offer more insights into the limitations of LLMs’ self-evaluation in more complex scenarios.
> >
> > We thank the reviewer for this insightful suggestion. To address this, we conducted a thorough investigation into the ability of LLMs to detect and rectify mistakes in their own reasoning steps. Specifically, we selected questions from the proposed datasets where the corresponding model produced an incorrect final answer, thereby indicating errors in the reasoning chain. These incorrect reasoning steps, generated by the same model, were used to evaluate its mistake detection, rectification, and ability to derive the correct answer.
> >
> > Our findings reveal that while models, such as GPT-4 and GPT-4O, demonstrate comparable performance in detecting mistakes in their own reasoning as they do with SLM-generated mistakes, their performance in rectifying errors and deriving correct final answers significantly drops when handling self-generated errors.
> >
> > For instance, using GPT-4 on the MATH dataset with 100 incorrect self-generated reasoning steps, we observed the following results for self-generated incorrect reasoning compared to SLM-generated reasoning:
> >
> > *   **Mistake Identification:** 0.914 (self-generated) vs. 0.90 (SLM-generated)
> > *   **Final Answer Accuracy:** 0.471 (self-generated) vs. 0.65 (SLM-generated)
> > *   **Rectification Performance:** 0.533 (self-generated) vs. 0.70 (SLM-generated)
> >
> > These findings suggest that while models can effectively identify mistakes in their own reasoning, the challenge lies in rectifying these errors and producing accurate final answers. This discrepancy underscores the limitations of LLMs in self-evaluation, particularly in more complex scenarios.
> >
> > We will include these detailed results across all datasets in the revised paper to provide further insights into the limitations and challenges of LLMs’ self-evaluation capabilities.
> >
> > ## Questions
> >
> > >Figure 2 is not referenced anywhere in the paper.
> >
> > Figure 2 shows an example from the curated MWP-dataset. This was an oversight and should be referenced in Section 2. We will rectify it.
> >
> > >Lines 208-209: Do you mean Table 2? What is the difference between Table 2 and Table 7?
> >
> > Thank you for catching this oversight. The reference on Lines 208–209 should indeed be to Table 2. Additionally, Tables 2 and 7 are duplicates, included both in the main paper and the appendix. We will remove the redundant table in the appendix in the revised manuscript.
> >
> > >Lines 242-243: Why is GPT-4o underperforming in both simpler and more complex datasets? A brief explanation of the possible challenges it faces with these datasets would be helpful.
> >
> > Please see general comment for the summary of al insights.
> >
> > >A detailed analysis of mistake types would strengthen the study. For instance, it would be valuable to know which types of mistakes models consistently struggle to detect.
> >
> > A detailed analysis of mistake types is already provided in Appendix F, where we break down mistake identification and performance of GPT-4O across GSM8K, MATH, MATHBENCH, and JEEBENCH datasets. Key observations include:
> >
> > 1.  Reasoning chains without actual mistakes are also constantly flagged as erroneous by the model.
> > 2.  The model particularly struggles with identifying errors in shuffle reasoning steps and delete reasoning steps. We believe these are critical errors in the reasoning chain that significantly impact logical coherence and should not be overlooked.
> >
> > We will ensure this analysis is clearly referenced and emphasized in the main paper to address this concern.

---

> > > ### Author Response · Authors · 2024-11-23
> > >
> > > Thank you again for your review. We have addressed all the concerns raised. Please let us know if you have any further questions. We are looking forward to a positive response and score.

---

> > > > ### Author Response · Authors · 2024-11-25
> > > >
> > > > Thank you again for your feedback and thoughtful questions! As we are now approaching the end of the rebuttal period, I’m following up again to check if you have any further questions regarding our responses. We hope we have addressed your concerns and clarifications thoroughly and looking forward to a positive response and score.

---

> > > > > ### Author Response · Authors · 2024-11-25
> > > > >
> > > > > Thank you again for your feedback and thoughtful questions! As we are now approaching the end of the rebuttal period, I’m following up again to check if you have any further questions regarding our responses. We hope we have addressed your concerns and clarifications thoroughly and looking forward to a positive response and score.

---

> > > ### Comment · Reviewer_ix9B · 2024-11-26
> > >
> > > Thanks for your response, especially the self-generated mistake detection. It seems that detecting reasoning errors from SLMs is quite easy for current LLMs. So I believe it is better to include more hard problems that really bottleneck current LLMs. Overall, I will keep my score.

---

> > > > ### Author Response · Authors · 2024-11-26
> > > >
> > > > Dear Reviewer,
> > > >
> > > > Thank you for your feedback and for acknowledging the challenges around SLM-generated mistake detection. As highlighted, errors from SLM-generated reasoning often cascade through multiple steps, making them relatively easier for models to detect. However, even with such seemingly simpler error chains, models exhibit poor generalization, especially on more complex datasets like JEEBench or MATHBench, where their ability to detect mistakes drops significantly.
> > > >
> > > > Generating step-by-step reasoning with harder, human-like mistakes is indeed challenging and requires considerable human effort. To address this, we introduced rule-based injected errors, which simulate authentic human mistakes localized to specific reasoning steps. These errors have proven to be much harder for models to detect, underscoring their limitations in identifying nuanced reasoning flaws.
> > > >
> > > > In summary, while models may perform better on simpler datasets with SLM-generated errors, they struggle significantly with both complex error reasoning chains and challenging datasets. This highlights a critical bottleneck in their reasoning capabilities, which we hope our dataset and analysis can help address.
> > > >
> > > > If you have further questions or suggestions, please let us know. We also emphasize that MWP-MISTAKE is one of the first datasets to comprehensively explore reasoning chain errors, offering valuable insights for building better datasets and advancing reasoning capabilities in models.

---

### Author Response · Authors · 2024-11-21

We appreciate the thoughtful feedback from all reviewers and the opportunity to address their concerns.

We present **MWP-MISTAKE dataset**, a human-verified dataset of math word problems with correct and incorrect reasoning chains. It evaluates state-of-the-art (SOTA) LLMs and SLMs on their ability to detect and correct reasoning mistakes. This paper presents a **comprehensive and novel evaluation framework** for reasoning mistakes in LLMs using MWP-MISTAKE. It provides critical insights into detection, correction, contamination, and interdependencies between tasks, all while addressing a timely and underexplored research problem. We believe this work is a significant step forward for advancing reasoning capabilities in LLMs.

**Addressing Reviewer Concerns:** Most concerns raised by reviewers revolved around clarification of experimental setups, metrics, and qualitative analyses. We have provided detailed explanations and will incorporate additional results, such as qualitative case studies, extended contamination experiments, and clarification of metrics in the revised manuscript. None of the concerns undermine the novelty, scope, or contributions of our work.  **We look forward to continued discussion and feedback from the reviewers and area chairs.**

1. **Timely Dataset and Problem**:  The reasoning capabilities of LLMs remain a bottleneck in their adoption for high-stakes tasks requiring mathematical or logical precision. MWP-MISTAKE addresses this gap by providing a high-quality dataset designed to evaluate and benchmark reasoning abilities. This aligns with current research trends, including the release of models like OpenAI's O1, which claim improved reasoning but still exhibit critical limitations in mistake detection, as highlighted by our findings.

2. **Comprehensive Evaluation of Reasoning Abilities**: Our evaluations uncover several key insights:

*   SOTA models frequently flag correct reasoning chains as erroneous, casting doubt on their reliability.
*   Models struggle more with localized, subtle errors (e.g., shuffle and delete reasoning steps) compared to propagated mistakes from SLMs.
*   Mistake detection directly influences the model's ability to rectify errors, revealing an interdependence between these tasks.

3. **Data Contamination and Memorization Challenges**: Data contamination and memorization remain a fundamental challenge in LLM research, especially with black-box models like GPT, Claude, and Gemini, where training data is inaccessible. Our findings show that datasets like GSM8K and MATH have significant contamination, which inflates model performance metrics artificially. Conversely, newer datasets such as JEEBENCH reveal genuine reasoning weaknesses, underscoring the importance of our work for creating contamination-free benchmarks.

4.  **Practical Contributions to Model Evaluation**:

*   Our findings show that models like GPT-4o and O1 achieve strong performance on answering but struggle to reliably detect errors in reasoning chains.

*   The paper introduces qualitative analyses and error categorization (e.g., rule-based versus SLM-generated mistakes) to understand why models fail on certain errors.

*    These insights can guide future research in building models with improved reasoning frameworks.

---

### Meta-Review · Area_Chair_QDGC · 2024-12-23

**Metareview:**

The paper presents a dataset for evaluating LLMs’ abilities in detecting and correcting mathematical reasoning mistakes. The dataset contains math word problems with both correct and wrong reasoning chains. The proposed pipeline includes perturbations to the original reasoning chains based on (1) rule-based techniques, and (2) using less capable language models.

The reviewers generally found the problem studied to be important, the presentation to be clear, the dataset to be helpful for the community, and that the experiments insightful.

However, reviewers found several issues regarding the paper, including:
-The scope of the contributions of the work to be limited and incremental with respect to existing findings in the literature
-The perturbation based method might not accurately reflect the mistakes made by human (partially addressed in rebuttal)
-Some of the relevant work might be overlooked
-Evaluation setup needs better clarity
-While the paper discusses several challenges such as contamination there are limited discussions or practical solutions discussed

While the author response attempts to address several of these issues, key concerns—such as the overall incremental contribution—remain unresolved.

**Additional Comments On Reviewer Discussion:**

The authors have made an effort to address most of the questions and comments raised. While I will not detail every response, I will focus on the major points. Overall, the authors have carefully addressed concerns regarding the clarification of experimental setups, analyses, and metrics. They have also argued for the paper's significance relative to prior work on evaluating LLMs and their mistakes, emphasizing the key insights presented.
Despite these discussions and improvements, reviewers' general impressions of the work remains largely unchanged.

---

### Decision · Program_Chairs · 2025-01-22

Reject